# Tumour-vasculature development via endothelial-to-mesenchymal transition after radiotherapy controls CD44v6+ cancer cell and macrophage polarization

Seo-Hyun Choi[1,4], A-Ram Kim[1], Jae-Kyung Nam[1], Jin-Mo Kim[2], Jee-Youn Kim[2], Haeng Ran Seo[3], Hae-June Lee[1], Jaeho Cho [2] & Yoon-Jin Lee [1]

It remains controversial whether targeting tumour vasculature can improve radiotherapeutic efficacy. We report that radiation-induced endothelial-to-mesenchymal transition (EndMT) leads to tumour vasculature with abnormal SMA+NG2+ pericyte recruitment during tumour regrowth after radiotherapy. *Trp53* (but not *Tgfbr2*) deletion in endothelial cells (ECs) inhibited radiation-induced EndMT, reducing tumour regrowth and metastases with a high CD44v6+ cancer-stem-cell (CSC) content after radiotherapy. Osteopontin, an EndMT-related angiocrine factor suppressed by EC-*Trp53* deletion, stimulated proliferation in dormant CD44v6+ cells in severely hypoxic regions after radiation. Radiation-induced EndMT significantly regulated tumour-associated macrophage (TAM) polarization. CXCR4 upregulation in radioresistant tumour ECs was highly associated with SDF-1+ TAM recruitment and M2 polarization of TAMs, which was suppressed by *Trp53* deletion. These EndMT-related phenomena were also observed in irradiated human lung cancer tissues. Our findings suggest that targeting tumour EndMT might enhance radiotherapy efficacy by inhibiting the re-activation of dormant hypoxic CSCs and promoting anti-tumour immune responses.

[1] Division of Radiation Biomedical Research, Korea Institute of Radiological & Medical Sciences, Seoul 139-706, Korea. [2] Department of Radiation Oncology, Yonsei University College of Medicine, Seoul 120-752, Korea. [3] Cancer Biology Research Laboratory, Institute Pasteur Korea, Gyeonggi-do 13488, Korea. [4] Present address: Department of Surgery, Memorial Sloan Kettering Cancer Center, New York, NY 10065, USA. These authors contributed equally: Seo-Hyun Choi, A-Ram Kim, Jae-Kyung Nam. Correspondence and requests for materials should be addressed to Y.-J.L. (email: yjlee8@kirams.re.kr)

Despite recent technological advances in radiotherapy, challenges relating to tumour targeting, dose limitations, and tumour relapse and escape remain. Multiple strategies for targeting cancer cells, cancer stem cells (CSCs), tumour stroma, and tumour endothelial cells (ECs), as well as improving anti-tumour immune responses to increase tumour radiosensitivity, are being developed[1–3].

Anti-angiogenic or vascular-destructive agents potentially enhance tumour responses to radiotherapy[4]. Several anti-angiogenics have been clinically evaluated in combination with radiotherapy[5,6]; however, their benefits are controversial. Bone marrow-derived cell (BMDC) recruitment to irradiated tumours may contribute to tumour relapse via vasculogenesis[7,8]. Although tumour-vasculature development after radiotherapy is not well characterized, targeting tumour ECs enhances radiotherapeutic efficacy; ceramide, sphingomyelinase, and Bax regulate EC apoptosis after irradiation[9,10]. Vascular damage may affect tumour responses to high radiation doses, e.g., during stereotactic radiosurgery/radiotherapy[11,12]. ECs lacking ataxia-telangiectasia mutated showed increased radiosensitivity[13]. However, it remains debatable whether EC targeting can improve radiotherapy efficacy.

Cancer cells that acquire radioresistance exhibit CSC-like characteristics[1,14]. CSCs are often quiescent after radiation or chemotherapy and their awakening causes tumour relapse and escape[15,16]. Understanding the mechanism regulating the dormant or proliferative status better is important for targeting CSCs.

Radiotherapy can stimulate anti-tumour immune responses. Immunomodulation using antibodies against programmed death 1 and programmed death-ligand 1 in combination with radiotherapy has been assessed in clinical trials[17]. Radiotherapy can enhance immunosuppressive responses, including chemotactic signals that recruit several myeloid cell types[17]. Radio-immunomodulation studies have revealed crucial strategies for effectively combining immunotherapy and radiotherapy.

Endothelial-to-mesenchymal transition (EndMT) promotes cancer-associated fibroblast formation in tumours[18], affects the endothelium to enable tumour-cell extravasation[19], and may give rise to pericyte-like cells within tumours[20]. Pericytes play critical roles in blood-vessel maturation and blood-barrier maintenance and regulate vessel integrity and function by interacting with ECs[21,22]. Tumour vessels harbouring less pericytes are more sensitive to radiation and chemotherapy[20,23].

Here, we studied tumour EndMT and pericyte-derived tumour vasculature during tumour regrowth after radiotherapy. We analysed the effects of EndMT-regulated vasculature on the irradiated tumour microenvironment, especially, hypoxic dormant CSCs and tumour-associated macrophage (TAM) polarization of bone marrow-derived monocytes (BMDMs).

## Results

**Trp53 and Tgfbr2 conversely regulate EndMT in vitro.** We previously reported radiation-induced EndMT in several EC types[24–26]. Trp53 is a key regulator of radiation responses in ECs, and tansforming growth factor-β (TGFβ)-related signalling potentially is a key regulator of EndMT[27,28]. Thus, we explored the effects of small interfering RNA (siRNA)-mediated knockdown of Trp53 and Tgfbr2 on radiation-induced EndMT in human umbilical vein ECs (HUVECs). At 48 h post irradiation (hpi), Trp53 silencing in HUVECs markedly inhibited irradiation-induced messenger RNA (mRNA) expression of Snail1, Snail2, and Zeb2, which encode transcription factors implicated in EndMT[29], compared to control siRNA-treated cells, whereas Tgfbr2 knockdown increased their expression

(Supplementary Fig. 1a, b). Accordingly, overexpression of Trp53, but not Tgfbr2, augmented irradiation-induced increases in the EndMT markers filamentous actin, vimentin, and smooth muscle actin (SMA), while reversing irradiation inhibited CD31 levels (Supplementary Fig. 1c, d). Pericytes significantly restored irradiation-induced tubule formation impairment, but not upon Trp53 knockdown, which inhibited pericyte recruitment (Supplementary Fig. 1e). In contrast, Tgfbr2 knockdown significantly enhanced pericyte integration into irradiated EC complexes and recovered EC tubule formation (Supplementary Fig. 1e).

**EC-Trp53 KO inhibits EndMT-related abnormal vasculature.** Inspired by our findings in vitro, we next analysed tumour-vasculature development during regression and regrowth after radiotherapy in syngeneic mouse tumours of colon carcinoma cells (CT26). The changes in tumour size are shown in Supplementary Fig. 2a. Irradiation significantly increased collagen deposition, especially around tumour vessels, during regression and regrowth, and CD31$^+$ areas (indicative of EC) and vessels were more dilated than in non-irradiated tumours (Supplementary Fig. 2b, c). The SMA$^+$CD31$^+$ population was significantly increased around hypoxic regions and was labelled with pimonidazole during tumour regression and regrowth (Supplementary Fig. 2d, e).

To study the potential relationship between tumour vasculature and radioresistance, we used EC-specific Trp53-knockdown (EC-p53KD) and -knockout (EC-p53KO) mice (Tie2-Cre; Trp53$^{flox/+}$ and Tie2-Cre;Trp53$^{flox/flox}$, respectively). A syngeneic tumour model was established by implanting tumour cells (referred to as KP cells) isolated from a spontaneous lung adenocarcinoma in conditional Kras$^{G12D}$;Trp53$^{flox/flox}$ mice[30]. Primary KP cells were implanted at passage 4 or less to maintain the cellular characteristics of spontaneous lung tumour. Trp53 mRNA$^-$VE-cadherin$^+$ cells were dominant in EC-p53KO, but not wild-type (WT) tumours, indicating that p53 was successfully knocked out in tumour ECs of EC-p53KO mice (Supplementary Fig. 3a).

KP tumours in WT, EC-p53KD, and EC-p53KO mice reached 150 mm$^3$ within comparable periods (Fig. 1a). However, following 20 Gy irradiation, tumour growth was significantly delayed in EC-p53KD/KO compared with WT mice (Fig. 1b, c). At 7 days post irradiation (dpi), necrotic areas and the apoptotic cell population were increased more in p53KO than in WT tumours (Fig. 1d, Supplementary Fig. 3b). Immunofluorescence analyses revealed that SMA$^+$CD31$^+$ lesions were significantly larger in peri- and intratumoural regions in irradiated (day 23) than in non-irradiated WT (day 15), but not EC-p53KO mice (Fig. 1e). CD31 and SMA colocalized (indicating EndMT) in WT, but not EC-p53KO ECs, on 3 and 23 dpi (Fig. 1f, Supplementary Fig. 3c). CD31$^+$ areas and vessel diameters were prominently larger during tumour regrowth post irradiation than in non-irradiated tumours (Supplementary Fig. 3d). SMA was highly expressed post irradiation in most tumour cells (besides ECs) in WT, whereas this was less evident in EC-p53KO mice (Fig. 1f, Supplementary Fig. 3e). Concurrently with increased SMA expression, collagen deposition increased significantly after irradiation in WT, but not EC-p53KD/KO mice (Supplementary Fig. 3f).

Many WT tumour vessels were covered with SMA$^+$NG2$^+$ pericytes 23 dpi compared with non-irradiated tumours in WT, but not EC-p53KO mice (Fig. 1g, h). Vessels with SMA$^+$NG2$^+$ pericytes appeared at 3 dpi, and most NG2$^+$ pericytes colocalized with SMA in irradiated WT tumour vessels (Supplementary Fig. 4a), concomitantly with radiation-induced EndMT. Desmin, another pericyte marker[31], was detected in irradiated WT tumours but did not colocalize with NG2$^+$ cells (Supplementary

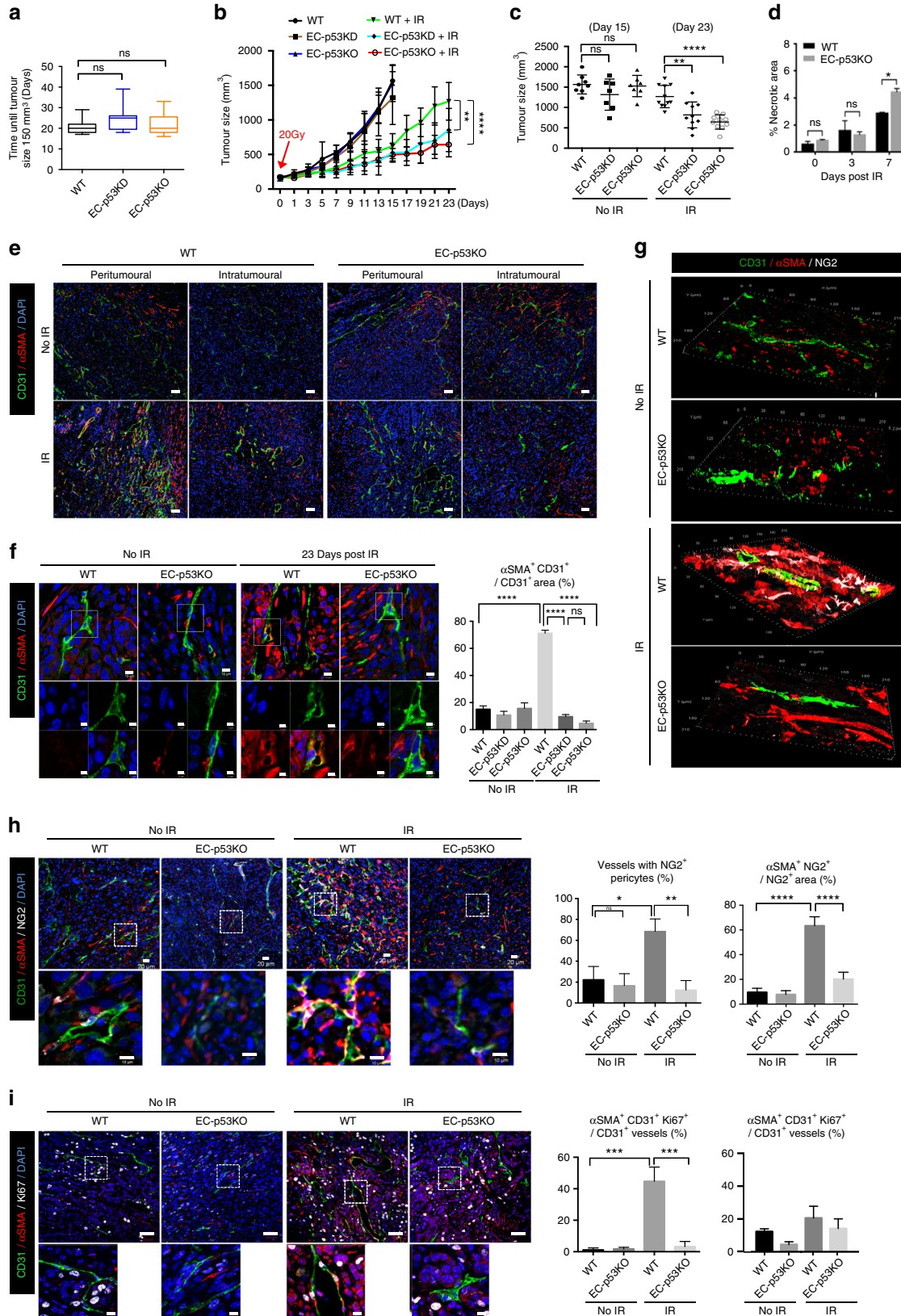

Fig. 4b). We analysed the EndMT and pericyte population in greater detail by endothelial lineage tracking using cells expressing tdTomato-labelled Cre. To this end, we generated EC-tdTomato and EC-tdTomato;p53KO mice (Tie2-Cre;tdTomato and Tie2-Cre;tdTomato Trp53$^{flox/flox}$, respectively)

(Supplementary Fig. 4c, left panel). tdTomato$^+$SMA$^+$ cells were significantly increased (>50%) at 7 dpi in tumours of EC-tdTomato, but not EC-tdTomato-p53KO mice (Supplementary Fig. 4c, middle-top and right panel). However, tdTomato$^+$NG2$^+$ cells were not detected in EC-tdTomato mice (Supplementary

**Fig. 1** EC-*Trp53* deletion inhibits EndMT, vascular abnormality, and KP tumour regrowth after irradiation. KP lung tumour cells were subcutaneously injected into the right hind legs of WT, *Tie2*-Cre;*Trp53$^{flox/+}$*(EC-p53KD), and *Tie2*-Cre;*Trp53$^{flox/flox}$* (EC-p53KO) mice. Tumour tissues (150–200 mm$^3$) were obtained from non-irradiated animals (days 0, 15) and after 20 Gy irradiation (IR; days 3, 23). Unless otherwise indicated, the numbers of animals in the respective groups were as follows: WT, $n = 8$; EC-p53KD, $n = 7$; EC-p53KO, $n = 7$; WT mice+IR, $n = 11$; EC-p53KD+IR, $n = 10$; EC-p53KO+IR, $n = 10$. **a** Days required for tumours to reach 150 mm$^3$ (WT, $n = 19$; EC-p53KD, $n = 17$; EC-p53KO, $n = 17$). **b** Tumour growth with or without irradiation. **c** Tumour sizes on days 15 (no irradiation) and 23 (after irradiation). **d** Quantification of the necrotic area per field (magnification, ×100) using H&E-stained images of tumours from WT and EC-p53KO mice, with or without irradiation (3 and 7 dpi). **e**, **f** Immunofluorescence of CD31 and αSMA in tumours on 0 and 23 dpi (**e**; Scale bar = 50 μm, **f**; Scale bar = 10 μm (crop, 5 μm)). Quantification of the αSMA$^+$CD31$^+$/CD31$^+$ area as an average of 5 fields (magnification, ×100) in (**d**) (no IR, $n = 7$; IR, $n = 10$). **g**, **h** Immunofluorescence of CD31, αSMA, and NG2 in tumours, with or without irradiation (23 dpi) and three-dimensional (3D) image reconstruction (**g**). Quantification of CD31$^+$ with NG2$^+$ pericyte coverage in the total CD31$^+$ area and αSMA$^+$NG2$^+$ cells in the total NG2$^+$ area (magnification, ×100; $n \geq 5$) from (**h**). Immunofluorescence of CD31, αSMA, and Ki67 in tumours from WT and EC-p53KO at 23 dpi. **i** The percentage of αSMA$^+$CD31$^+$Ki67$^+$ and αSMA$^-$CD31$^+$Ki67$^+$ vessels among all CD31$^+$ vessels per field (magnification, ×100; $n \geq 5$). Scale bar = 20 μm (crop, 10 μm). For (**a**–**c**), error bars indicate SD. For (**e**–**i**), error bars indicate SEM; *$p < 0.05$, **$p < 0.01$, ***$p < 0.001$, and ****$p < 0.0001$, ns not significant (one-way ANOVA for multiple comparison). Data are representative of three independent experiments

Fig. 4c, middle-bottom panel). During post-irradiation tumour regrowth, SMA$^+$ mesenchymal cell or NG2$^+$SMA$^+$ pericyte populations significantly increased around irradiated vessels. Based on these data, we cautiously suggest that NG2$^+$SMA$^+$ cells do not originate from ECs, whereas SMA$^+$ mesenchymal cells can be derived from ECs via EndMT.

EndMT (SMA$^+$CD31$^+$) cells in WT tumours were less TUNEL$^-$ or γH2AX$^-$ at 3 dpi, but highly Ki67$^+$ (indicative of cell proliferation) during tumour regrowth (23 dpi), in contrast to SMA$^-$ ECs in EC-p53KO mice (Fig. 1i and Supplementary Fig. 4d, e). To evaluate whether irradiated tumour EndMT vessels were functional, mice were intravenously injected with Hoechst 33342. Hoechst 33342 accumulated significantly more around EndMT-associated vessels in WT than in EC-p53KO tumours upon irradiation (Supplementary Fig. 4f). Further, significant fluorescein isothiocyanate–dextran leakage (indicative of intratumoural leakage) was observed in irradiated EC-p53KO compared to irradiated WT tumours (Supplementary Fig. 4g).

**EC-*Tgfbr2* knockdown accelerates EndMT and tumour growth**. TGFβ signalling is important in tumour responses to radiotherapy[32,33] and regulates EndMT[28]. After irradiation or TGFβ1 treatment, p-SMAD2/3 levels did not decrease in TGFβR1-depleted HUVECs, but they did increase upon TGFβR2 depletion (Supplementary Fig. 5a, b). Radiation- or TGFβ1-induced increases in EndMT markers (filamentous actin and FSP1) were decreased in TGFβR1-depleted HUVECs, but were increased in TGFβR2-depleted cells (Supplementary Fig. 5c, d). Therefore, we studied the relationship between radiation-induced tumour EndMT and tumour-irradiation responses in EC-specific *Tgfbr2*-knockdown (EC-TGFβR2KD) mice (*Tie2*-Cre;*Tgfbr2$^{flox/+}$*). Supplementary Figure 5e shows a non-irradiated tumour with EC-specific *Tgfbr2* knockdown. Post-irradiation p-SMAD2/3 levels of CD31$^+$ vessels were significantly higher in EC-TGFβR2KD than in WT mice (Supplementary Fig. 5f). In syngeneic KP tumours, *Tgfbr2* knockdown did not affect non-irradiated tumour growth (Fig. 2a), but it significantly enhanced tumour growth (Fig. 2b, c) and radiation-induced EndMT (Fig. 2d) post irradiation. Concurrently, *Tgfbr2* knockdown significantly increased collagen deposition in irradiated vs. non-irradiated tumours (Supplementary Fig. 5g). CD31 and SMA colocalized from 1 dpi in EC-TGFβR2KD mice (Supplementary Fig. 5h). Consistently, CD31$^+$ tumour vessel areas with NG2$^+$ pericytes were significantly larger in EC-TGFβR2KD than in WT mice (Fig. 2e). We examined γH2AX$^+$ and TUNEL$^+$ ECs shortly after irradiation to evaluate early vessel responses. SMA$^+$ vessels were more TUNEL$^+$ in WT than in EC-TGFβR2KD tumours at 3 dpi, but no difference in γH2AX was evident (Supplementary

Fig. 5i, j). The different tumour growth rates and pericyte populations in WT tumours in Fig. 2b–e vs. Fig. 1b–h may reflect different proliferation rates in KP cells at 10 dpi (Supplementary Fig. 5k); after irradiation, the proliferation rate of the primary KP cells used in the tumour-growth experiment shown in Fig. 2b was lower than that of the KP cells represented in Fig. 1b (Supplementary Fig. 5k).

**EC-*Trp53* KO overcomes *Tgfbr2*-deleted radioresistance**. To test whether *Trp53*-knockdown or -knockout could overcome the increased radioresistance of EC-TGFβR2KD tumours, we generated EC-p53KD;TGFβR2KD and EC-p53KO;TGFβR2KD mice. These mice grew to adulthood normally, and KP tumour growth to 150 mm$^3$ was unaffected (Fig. 2f). Following 20 Gy irradiation, tumour growth was significantly delayed in EC-p53KD/KO; TGFβR2KD vs. EC-TGFβR2KD mice (Fig. 2g, h). Vessel numbers with SMA$^+$NG2$^+$ pericytes decreased markedly after irradiation in EC-p53KO;TGFβR2KD vs. EC-TGFβR2KD tumours, but increased in TGFβR2KD vs. WT tumours (Fig. 2i).

Increasing evidence implicates BMDCs in vessel formation, especially vasculogenesis, in irradiated tumours[7]. To examine the relationship between BMDCs and tumour EndMT, we analysed tumour regrowth following radiotherapy in CT26 tumour-bearing mice after transplantation of CM-Dil (cell tracker)-labelled BMDCs (Fig. 2j). BMDCs did not colocalize with CD31$^+$ cells during tumour regrowth, but showed a pericyte-like phenotype around ECs (Fig. 2j). Tumour vessels in WT and EC-p53KO mice that had received BMDCs from *Tie2*-GFP mice did not colocalize with GFP$^+$ cells (endothelial progenitor cells) during tumour regrowth after radiotherapy (Fig. 2k). Indeed, vessels around recurrent lung tumours did not show EndMT (Supplementary Fig. 6).

**EC-*Trp53* KO affects hypoxic CD44v6$^+$ cell proliferation**. In radioresistant CSCs during tumour regrowth, aldehyde dehydrogenase$^+$, CD44$^+$, CD133$^+$, and epithelial adhesion molecule$^+$ lesion areas increased by 11%, 19%, 24%, and 4% respectively, in irradiated vs. control tumours, which showed no significant difference compared to irradiated EC-p53KO tumours (Supplementary Fig. 7a, b). However, CD44v6$^+$ areas increased by >50% in WT tumours but remained at 29% in EC-p53KO tumours after radiotherapy (Fig. 3a, Supplementary Fig. 7a). The CD44v6$^+$ cell population was the most strongly increased after radiotherapy. We hypothesize that CD44v6 may be the only tested CSC marker affected by radiation-induced tumour EndMT. Using green fluorescent protein (GFP)-overexpressing CT26 cells, we verified that mainly CD44v6 was increased in GFP$^+$ cell membranes during tumour regression and regrowth when compared to non-irradiated

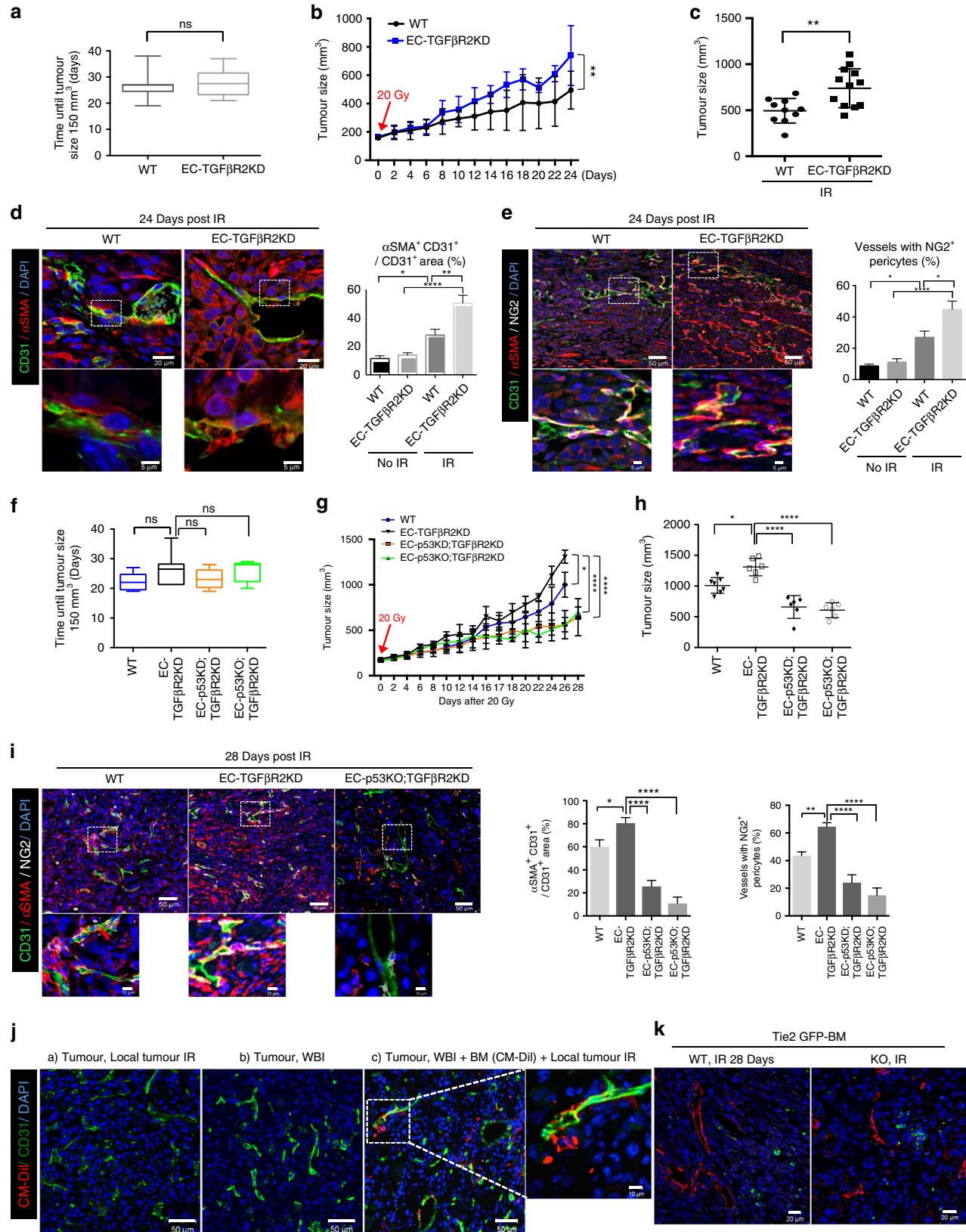

tumours (Supplementary Fig. 7c). Next, we studied whether this CD44v6$^+$ population increase resulted from CD44v6$^-$ primary KP cells that became CD44v6$^+$ after irradiation or was derived from proliferating radioresistant CD44v6$^+$ cells. CD44v6$^+$ cell numbers significantly increased at 3 and 7 dpi in WT, but not EC-p53KO tumours (Supplementary Fig. 7c).

tumours (Supplementary Fig. 7d). CD44v6$^+$ cells in WT tumours were TUNEL$^-$ at 3 dpi, and the TUNEL$^+$-to-total tumour area ratio was only 2.3% (Supplementary Fig. 7d). Indeed, in primary KP cells, the CD44v6$^+$ population increased in a dose-dependent manner at 24 hpi (Supplementary Fig. 7e).

**Fig. 2** EC-*Trp53;Tgfbr2* dual silencing overcomes tumour radioresistance associated with EC-*Tgfbr2* knockdown. **a–e** KP cells were injected subcutaneously into the right hind legs of WT and *Tie2*-Cre;*Tgfbr2*$^{flox/+}$ (EC-TGFβR2KD) mice. Tumour tissues were obtained before and 24 dpi (WT mice, $n = 12$; EC-TGFβR2KD mice, $n = 14$). **a** Days required for tumours to reach 150 mm$^3$ after KP cell injection. **b** Tumour growth following a single 20 Gy irradiation dose. **c** Tumour size at 24 dpi. **d** Immunofluorescence of CD31 and αSMA in tumours 24 dpi, with αSMA$^+$CD31$^+$ cell quantification (magnification, ×200; $n > 5$). Scale bar = 20 μm (crop, 5 μm). **e** Immunofluorescence of CD31, αSMA, and NG2 in irradiated tumours (24 dpi) and quantification of vessels with NG2$^+$ pericytes (magnification, ×200; $n > 5$). Scale bar = 50 μm (crop, 5 μm). **f–i** KP cells were injected subcutaneously into the right hind legs of WT, *Tie2*-Cre; *Trp53*$^{flox/+}$*Tgfbr2*$^{flox/+}$ (EC-p53KD;TGFβR2KD), and *Tie2*-Cre;*Trp53*$^{flox/flox}$*Tgfbr2*$^{flox/+}$ (EC-p53KO;TGFβR2KD) mice. Tumour tissues were obtained before and 28 dpi (WT mice, $n = 7$; EC-TGFβR2KD mice, $n = 6$; EC-p53KD;TGFβR2KD mice, $n = 6$; EC-p53KO;TGFβR2KD mice, $n = 6$). **f** Days required for tumours to reach 150 mm$^3$ after KP cell injection. **g** Tumour growth after 20 Gy irradiation in a single dose. **h** Tumour size on 28 dpi. **i** Immunofluorescence of CD31, αSMA, and NG2 in KP tumours 28 dpi, and αSMA$^+$CD31$^+$ vessels with NG2$^+$ pericytes quantification (magnification, ×200; $n > 5$). **j** Immunofluorescence of CM-Dil and CD31 in CT26 tumours from WT mice. WT mice were locally irradiated (Local IR), whole-body irradiated (WBI), or WBI and transplanted with WT mouse-derived BM (labelled with CM-Dil). Scale bar = 50 μm (crop, 10 μm). **k** GFP and CD31 in tumours from WT and EC-p53KO mice 28 dpi. WT or EC-p53KO mice were WBI and transplanted with *Tie-2*;GFP mouse-derived BM cells. Scale bar = 20 μm. The error bars indicate SD (**a–c**, **f–h**) or SEM (**d**, **e**, **i**); *$p < 0.05$, **$p < 0.01$, ****$p < 0.0001$, ns not significant (**a–c**: Student's *t*-test, **f–h**: one-way ANOVA for multiple comparison). All data are representative of three independent experiments

During tumour regrowth after 20 Gy irradiation, the pimonidazole-positive hypoxic area and staining intensity increased more in EC-p53KO than in WT tumours (Fig. 3b). However, in non-irradiated tumours, hypoxia was comparable between EC-p53KO and WT tumours (Fig. 3b, upper graphs). We hypothesize that EndMT inhibition in irradiated EC-p53KO tumours resulted in a loss of pericyte coverage and subsequent vascular leakage, resulting in increased tumour hypoxia.

CD44v6$^+$ tumour cells localized independently of the hypoxic region. Notably, CD44v6$^+$ cancer cells that localized in radiation-induced, extremely hypoxic regions of EC-p53KO tumours were Ki67$^-$ (no proliferation), in contrast to CD44v6$^+$ cancer cells in non-severely hypoxic areas of irradiated WT tumours (Fig. 3b, lower left graph). The number of Ki67$^+$ cells per tumour section was remarkably lower in EC-p53KO than in WT tumours, following radiotherapy (Fig. 3b, lower right graph). We hypothesize that several days after irradiation, vascular disruption in EC-p53KO tumours significantly increases, with subsequent formation of functionally defective vessels. Thus, severely hypoxic regions observed in EC-p53KO tumours may limit CD44v6$^+$ cell proliferation during tumour regrowth. Surprisingly, the number of KP metastatic nodules in the lungs was significantly lower in EC-p53KO than in WT mice after irradiation (Fig. 3c–e). Notably, in irradiated WT mice, >60% of cells in metastatic (*Kras*$^{G12D}$) lung tumour colonies (>100 μm tumour diameter) were CD44v6$^+$ vs. 25% in non-irradiated mice (Fig. 3e).

EC-*Tgfbr2* knockdown did not affect tumour hypoxia and CD44v6$^+$ cell proliferation in hypoxic regions after radiotherapy (Fig. 3f). Consistently, CD44v6$^+$ tumour cell numbers significantly more strongly increased 24 dpi in EC-TGFβR2KD vs. WT mice (Fig. 3f). The number of lung-metastatic tumour colonies was significantly increased in EC-TGFβR2KD vs. WT mice after irradiation, and CD44v6$^+$ cells were more prominent in irradiated than in non-irradiated metastatic tumours (Fig. 3g, Supplementary Fig. 8a). In agreement herewith, KP lung-metastatic nodule numbers were significantly decreased in EC-p53KD/KO;TGFβR2KD vs. WT mice after radiotherapy (Fig. 3h). Lung-metastatic tumour cells did not originate from NG2$^+$SMA$^+$ pericytes in irradiated tumours, indicating the important role of radiation-induced CD44v6$^+$ cells in metastasis (Supplementary Fig. 8b).

**EC-*Trp53* KO does not reduce tumour growth after fractionated radiotherapy.** To examine whether fractionated radiotherapy influences EndMT, we irradiated WT and EC-p53KO tumours with 30 Gy in six fractions (Supplementary Fig. 9a). We observed no difference in tumour growth and lung metastasis between WT and EC-p53KO (Supplementary Fig. 9b-d).

However, the fractionated irradiation-induced increase in the SMA$^+$CD31$^+$ population in WT tumours was significantly inhibited in EC-p53KO tumours, whereas pimonidazole-staining intensity and the Ki67$^+$CD44v6$^+$ cancer cell population in hypoxic areas were not different between WT and EC-p53KO tumours (Supplementary Fig. 9e, f).

In addition, we examined the effect of 20 Gy exposure in daily 2 Gy fractions in WT tumours and EC-p53KO mice established by systemic tamoxifen-mediated-specific Cre recombination in the VE-cadherin promoter (Supplementary Fig. 9g-k). After daily 2 Gy fractions, WT tumour growth was significantly higher than that observed with a single 20 Gy dose. However, EndMT occurrence and hypoxic staining density in WT tumours was significantly decreased after fractionated compared to a single high-dose irradiation (Supplementary Fig. 9j, k). Coincident with WT tumour growth, the proliferative CD44v6$^+$ cancer cell population in hypoxic areas increased more with 2 Gy fractions than with the 20 Gy dose (Supplementary Fig. 9k). Moreover, no difference in tumour growth and the Ki67$^+$CD44v6$^+$ population occurred in the hypoxic areas in WT and EC-p53KO mice after 10 daily 2 Gy fractions, even though EC-p53KO significantly reduced EndMT.

**EndMT-related Osteopontin stimulates dormant CD44v6$^+$ cells.** To elucidate whether TRP53 is critical for radiation-induced EndMT, we subjected HUVECs transfected with siRNAs against *Trp53*, *Tgfbr2*, or *Trp53*+*Tgfbr2* mRNA to RNA-sequencing (RNA-seq) analysis. Various EndMT progression-associated molecules were upregulated at 48 hpi (Fig. 4a). In TGFβR2-silenced ECs, these patterns were more pronounced, whereas in *Trp53*-deficient cells, the expression of genes promoting EndMT progression was significantly decreased, and TGFβR2-enhanced EndMT-related genes were downregulated (Fig. 4a). Quantitative reverse-transcription PCR (RT-qPCR) data indicated that EC-specific gene expression was not significantly changed, whereas expression of fibroblast-associated genes, including fibroblast-activation protein (FAP) and alpha smooth muscle actin (ACTA2), prominently increased during radiation-induced EndMT, indicating that TRP53 is a key regulator of radiation-induced EndMT (Fig. 4b). Moreover, irradiation downregulated EC-specific genes (*CD34*, *PECAM-1*, *CDH5*, and *KDR*) and upregulated fibroblast-specific genes (*COL1A1*, *COL6A1*, *ACTA2*, *FN1*, and *FAP*) in KP ECs isolated from WT, but not EC-p53KO mice (Supplementary Fig. 10a, b). In addition, *OPN* mRNA levels increased by >7-fold in control and TGFβR2-deficient ECs at 48 hpi, but were significantly suppressed by *Trp53* knockdown (Fig. 4c). Coincident with *OPN* upregulation, secreted OPN increased in control and

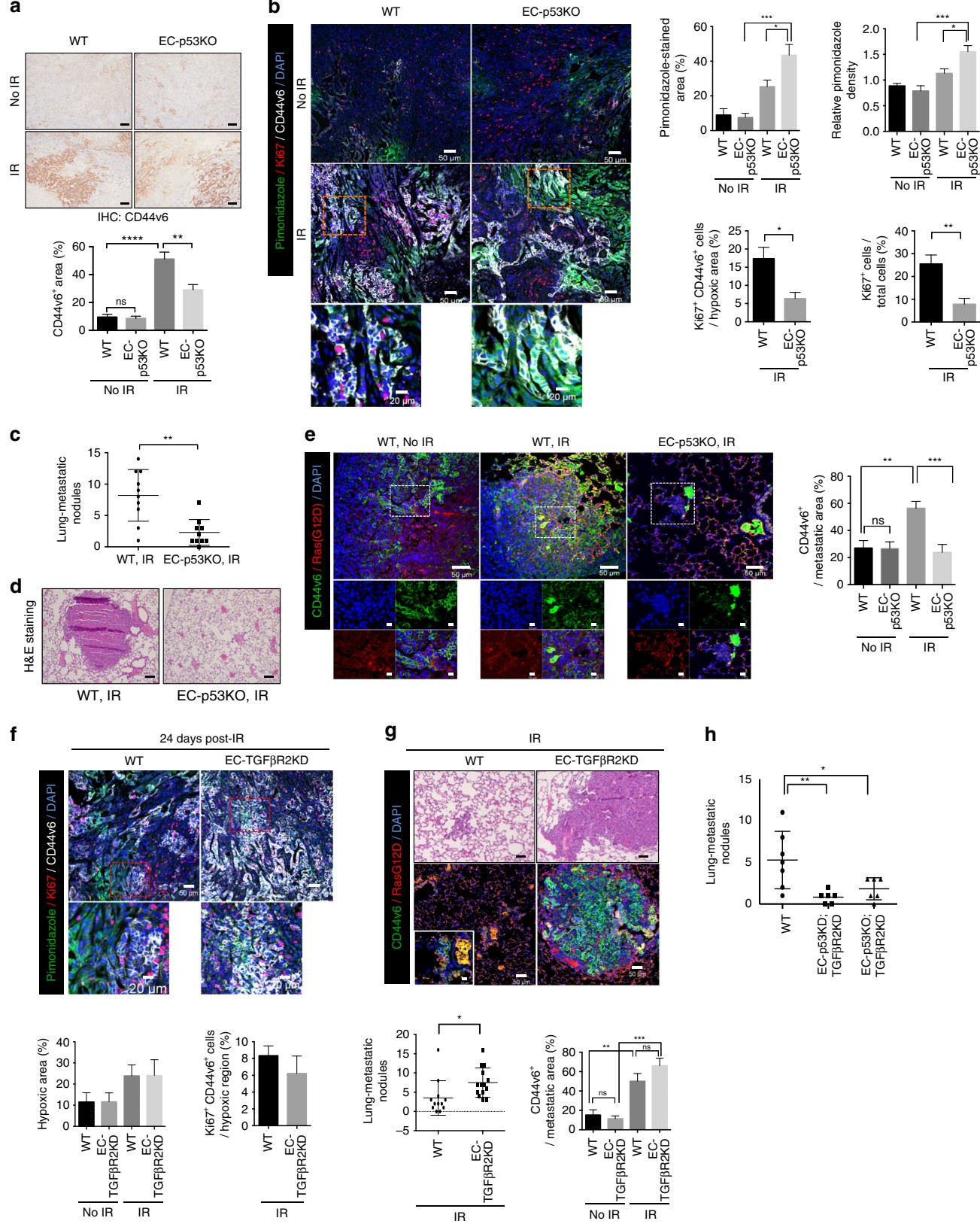

TGFβR2-deficient ECs at 5 dpi, but significantly decreased after *Trp53* knockdown in conditioned HUVEC medium (Supplementary Fig. 10c). Cytokine immunoassays with conditioned media confirmed that irradiated ECs secreted more OPN than did non-irradiated ECs (Fig. 4d).

Because OPN can interact with CD44[34], we analysed the relation between EC-secreted OPN and CD44v6+ CSCs in irradiated tumours. Surprisingly, SMA+ tumour ECs of irradiated WT, but not EC-p53KO mice highly expressed OPN (Fig. 4e, Supplementary Fig. 11a). Significant OPN secretion from vessels

**Fig. 3** Elevated proliferating CD44v6[+] cancer cells and lung metastasis after radiation are oppositely regulated by EC-*Trp53* and EC-*Tgfbr2*. **a** Immunohistochemistry of CD44v6[+] cells in KP tumours from WT and EC-p53KO mice, with or without irradiation (23 dpi). scale bars, 100 μm. Quantification of the CD44v6[+] area/field (magnification, ×200). **b** Immunofluorescence of pimonidazole, Ki67, and CD44v6 in WT and EC-p53KO tumours 23 dpi. Scale bar = 50 μm (crop, 20 μm). Quantification of pimonidazole-positive areas, relative pimonidazole densities, Ki67[+]CD44v6[+] cells in total hypoxic regions, and percentages of Ki67[+] cells/field (magnification, ×200) in (**b**). For all measurements, n > 5. **c** Quantification of lung-metastatic nodules/mouse from KP tumour-bearing mice at 23 dpi (n = 10). **d** H&E-stained lung sections (scale bars, 100 μm) from KP tumour-bearing mice 23 dpi. **e** Immunofluorescence of CD44v6[+] and active KRAS (KRAS[G12D]) in metastatic lung KP tumours with or without irradiation, and quantification of the CD44v6[+] area percentage in KRAS[G12D+] lung-metastatic tumours (magnification, ×200; n = 10). Scale bar = 50 μm (crop, 10 μm). **f** Immunofluorescence of pimonidazole, Ki67, and CD44v6 in KP tumours from irradiated WT and EC-TGFβR2KD mice (24 dpi); the percent hypoxic area/field; and the number of Ki67[+]CD44v6[+] cells/ hypoxic region (magnification, ×200, n > 5). Scale bar = 50 μm (crop, 20 μm). **g** Immunofluorescence of CD44v6 and KRAS[G12D] in lung-metastatic KP tumours 24 dpi, and the numbers of lung-metastatic nodules/mouse (WT mice, n = 12; EC-TGFβR2KD mice, n = 14). H&E-stained lung sections; scale bars, 100 μm, Quantification of the CD44v6[+] area/KRAS[G12D+] lung-metastatic tumour (magnification, ×200; n > 5). Scale bar = 50 μm (crop, 10 μm). **h** Quantification of lung-metastatic nodules/mouse from KP tumour-bearing WT (n = 7), EC-p53KD;TGFβR2KD (n = 6), and EC-p53KO; TGFβR2KD (n = 6) mice at 28 dpi. IR irradiation (20 Gy). For (**a**, **b**, **e**–**g**), error bars indicate SEM. For (**c**, **h**) error bars indicate SD; *p < 0.05, **p < 0.01, ***p < 0.001, ****p < 0.0001, ns not significant (**b** lower, **f** right and **g** left graphs: Student's t-test, others: one-way ANOVA). All data are representative of three independent experiments

that underwent EndMT occurred 7 dpi (Supplementary Fig. 11b). Notably, CD44v6[+] CSCs preferentially localized adjacent to OPN[+] ECs in irradiated WT tumours, but not in non-irradiated WT and irradiated EC-p53KO tumours (Fig. 4f, g).

In contrast, EC-*Tgfbr2* knockdown increased OPN[+]CD44v6[+] regions after radiotherapy (Supplementary Fig. 11c). To investigate the effects of OPN on dormant CD44v6[+] CSCs in severely hypoxic regions, we analysed the proliferative portion of irradiated CD44v6[+] CSCs under normoxic (20% $O_2$) and hypoxic (1%/0.5% $O_2$) conditions with/without OPN, in vitro. Hypoxia increased CD44v6 expression, with non-proliferative (Ki67[−]) KP cells surviving after irradiation (Fig. 4h). Fluorescence-activated cell sorting (FACS) analysis confirmed that under hypoxia, irradiated KP cells in the EdU[+] population showed significantly increased CD44v6 expression in the presence of OPN (Fig. 4i). Moreover, immunofluorescence data showed that nuclear expression of several stemness markers (Oct-4/Sox-2/β-catenin) increased substantially in the OPN-induced CD44v6[+] population under hypoxia, compared to irradiated KP cells (Supplementary Fig. 11d).

To delineate the role of OPN secretion during post-radiotherapy tumour regrowth, we examined the effects of a neutralizing anti-OPN antibody on EC-TGFβR2KD tumours. Notably, in EC-TGFβR2KD mice, the anti-OPN antibody reduced tumour growth after irradiation, compared to control IgG (Supplementary Fig. 12a). However, compared to control IgG, the anti-OPN antibody markedly reduced the formation of OPN[+]SMA[+] vessels (from >60 to 10%) and proliferative CD44v6[+] cells in EC-TGFβR2KD tumours at 21 dpi, and it induced similar effects in WT tumours (Supplementary Fig. 12b, c).

**EndMT modulates M1/M2 TAM populations after radiotherapy.** RNA-seq analysis was used to investigate the effects of TRP53-regulated EndMT on the tumour environment during post-irradiation tumour growth. Among genes coregulated in irradiated cells with/without TGFβR2 siRNA (indicated by an asterisk in Fig. 5a, left), genes adversely regulated by *Trp53* knockdown were considered for further analysis. Network analysis of matrisome- and surface-associated genes (Fig. 5a, right) indicated that enrichment for leukocyte migration- and extracellular organization-related genes strongly correlated with TRP53-regulated, radiation-induced EndMT (Fig. 5b).

Because radiation causes leukocyte recruitment, which likely regulates tumour responses to radiotherapy[35], we examined TAM patterns after radiotherapy. At 7 dpi, arginase 1 (Arg1)[+]F4/80[+] TAMs (M2-type) were highly accumulated in irradiated WT, but

not EC-p53KO tumour vessels, compared to non-irradiated control vessels (Fig. 5c, left). Radiation significantly increased the population of iNOS[+]F4/80[+] TAMs (M1-type) in WT tumours (Fig. 5c, right). The M1-type TAM population was two-fold higher in EC-p53KO than in WT tumours after radiation (Fig. 5c, d, upper panels). In contrast, radiation induced a stronger increase in M2-type TAMs in EC-TGFβR2KD than in WT tumours, whereas M1-type TAMs were lower in EC-TGFβR2KD than in WT tumours after radiation (Fig. 5d, lower panels and Supplementary Fig. 13a). FACS analysis confirmed that radiation increased the proportion of the F4/80[+]CD206[+] M2 to total TAMs by >2-fold in WT, but not EC-p53KO tumours vs. non-irradiated tumours in vivo (Fig. 5e).

To clarify whether radiation-induced tumour EndMT can directly affect M1/M2 populations, we cocultured non-irradiated or irradiated tumour ECs with macrophages differentiated from BMDMs by using macrophage colony-stimulating factor (M-CSF). When cocultured with irradiated WT, but not EC-p53KO tumour ECs showing EndMT, F4/80[+] macrophages showed an ~50% increase in the CD206[+] (or Arg1[+]) M2 subtype vs. macrophages cocultured with non-irradiated tumour ECs (Fig. 5f). In contrast, the iNOS[+] M1 population increased by >50% after coculture with irradiated p53KO, but not WT tumour ECs (Fig. 5f). Coculture with WT, but not EC-p53KO irradiated tumour ECs, increased F4/80[+] macrophage proliferation by >2.5-fold (Fig. 5f).

**EC-*Trp53* KO attenuates M2 polarization of SDF-1[+] TAMs.** The above RNA-seq analysis indicated that CXCR4 (SDF-1 receptor) is highly expressed on irradiated ECs, which is inhibited by Trp53 deficiency (Fig. 6a). Indeed, at 7 dpi, WT tumour ECs showed strongly increased CXCR4 expression compared to non-irradiated tumour ECs, whereas EC-p53KO tumours did not (Fig. 6b). Interestingly, at 1 dpi, SDF-1[+] cells were highly recruited to both irradiated WT and EC-p53KO tumours (Fig. 6c). SDF-1[+] cells often contacted tumour ECs, especially in irradiated WT tumours (Fig. 6d). Because blood monocytes secrete SDF-1, which promotes their differentiation into macrophages[36], we investigated the effects of monocytic SDF-1 recruitment and endothelial CXCR4 on radiotherapy outcome. At 7 dpi, >60% of SDF-1[+]F4/80[+] cells appeared to be CD206[+] M2 macrophages in WT, but not EC-p53KO tumours, whereas more iNOS[+] M1 macrophages were found in EC-p53KO than in WT tumours (Fig. 6e, Supplementary Fig. 13b). These M2 subtype patterns were confirmed by FACS analysis, which showed that most CD206[+]F4/80[+] cells were SDF-1[+] (Fig. 6f).

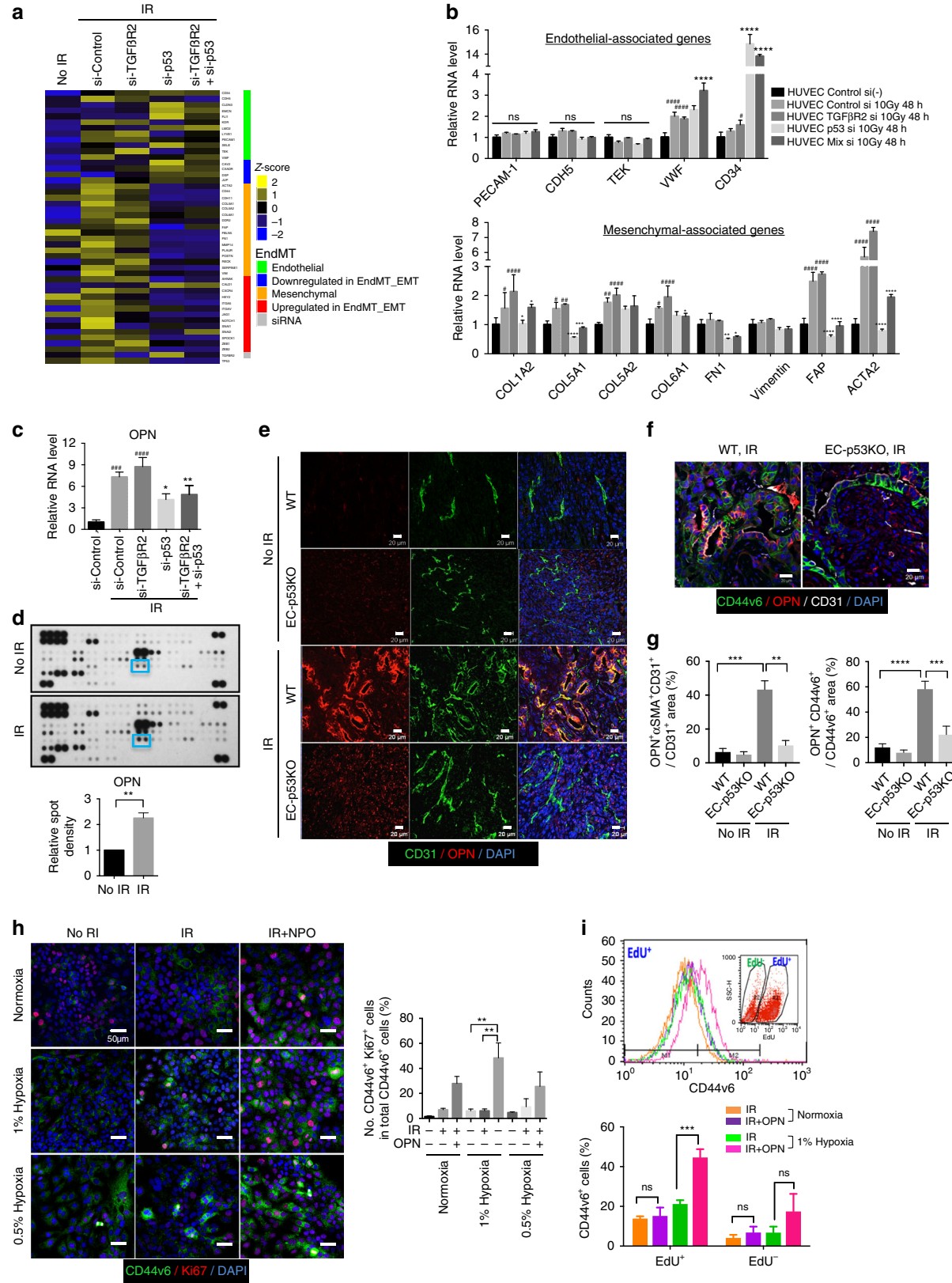

To ascertain whether TRP53-regulated tumour EndMT directly affects SDF-1$^+$ monocyte differentiation to M1/M2 TAMs, irradiated tumour ECs were cocultured for 48 h with macrophages differentiated from BMDMs by M-CSF. In accordance with the in vivo results, most F4/80$^+$SDF-1$^+$ cells were CD206$^+$

(M2) after coculture with WT tumour ECs, but iNOS$^+$ (M1) after coculture with EC-p53KO tumour ECs (Fig. 6g). Following coculture with WT tumour ECs, most CD206$^+$SDF-1$^+$F4/80$^+$ cells were elongated (like M2 macrophages) when differentiated with cytokines interleukin-4 (IL-4) and IL-10 from M-CSF-

**Fig. 4** Increased OPN expression during radiation-induced EndMT correlates with proliferating CD44v6$^+$ cancer cells in hypoxic areas. **a–c** HUVECs were transfected with siRNAs targeting *TGFβR2*, *Trp53*, or *TGFβR2+ Trp53* for 48 h and irradiated (10 Gy). After 48 h in culture, total RNA was isolated for RNA-seq and RT-qPCR analyses. **a** Heat map of RNA-seq analysis showing inhibition of radiation-induced EndMT and the mesenchymal phenotype in HUVECs by *Trp53* knockdown. **b** RT-qPCR analysis of EC-adhesion molecules, collagen, and fibroblastic markers in HUVECs. **c** RT-qPCR analysis of OPN in HUVECs. **d** Human soluble-receptor array analysis of conditioned medium from human pulmonary microvascular endothelial cells at 8 days after 5 Gy irradiation. The cyan boxes mark the spots corresponding to OPN. **e, f** Immunofluorescence detection of CD31 and OPN (**e**) or CD44v6, OPN, and CD31 (**f**) in KP tumours from irradiated WT and EC-p53KO mice (23 dpi). Scale bar = 20 μm. **g** Quantification of OPN$^+$αSMA$^+$CD31$^+$ cells in the total CD31$^+$ area (left) and OPN$^+$CD44v6$^+$ cells in the total CD44v6$^+$ area (right) (magnification, ×200; $n > 5$). Mouse strain designations are the same as in Figs 1 and 2. IR irradiation (20 Gy). **h, i** KP tumour cells were cultured under normoxia, 1% O$_2$ (mild hypoxia), or 0.5% O$_2$ (severe hypoxia) and irradiated with or without OPN. Data are representative of three independent experiments. **h** Immunofluorescence detection of CD44v6 and Ki67 in KP tumour cells 11 days after irradiation (left) and quantification of CD44v6$^+$Ki67$^+$ cells in total CD44v6$^+$ cells (right) (magnification, ×200; $n > 5$). Scale bar = 50 μm. **i** Flow-cytometric analysis of EdU incorporation and CD44v6 expression in KP tumour cells 11 days after irradiation. For (**b–d**), the error bars indicate SD of three independent experiments. For (**g–i**), the error bars indicate SEM; For (**b, c**), si-TGFβR2 + IR or si-Control + IR versus si-Control group, #$p < 0.05$, ##$p < 0.01$, ###$p < 0.001$, and ####$p < 0.0001$.si-p53 + IR or si-TGFβR2+si-p53 + IR versus si-Control + IR or si-TGFβR2 + IR group, respectively, *$p < 0.05$, **$p < 0.01$, ***$p < 0.001$, ****$p < 0.0001$, ns not significant (**d** Student's *t*-test, others: one-way ANOVA for multiple comparison)

---

treated BMDMs, whereas after coculture with EC-p53KO tumour ECs, iNOS$^+$SDF-1$^+$F4/80$^+$ cells were flattened (like M1 macrophages) when differentiated with lipopolysaccharides (LPS) plus interferon-γ (IFN-γ) from M-CSF-treated BMDMs (Fig. 6g).

We hypothesized that SDF-1$^+$ BMDMs are preferentially recruited by CXCR4-expressing ECs after radiotherapy. In vivo, treatment with the CXCR4 antagonist AMD3100 significantly reduced SDF-1$^+$CD206$^+$ macrophage accumulation around tumour vessels after radiotherapy, with increased iNOS$^+$ M1 macrophage and non-differentiated SDF-1$^+$ monocyte populations (Fig. 6h and Supplementary Fig. 13c). AMD3100 treatment did not affect CXCR4 upregulation after radiation (Supplementary Fig. 13d). Radiation increased the proliferation (BrdU$^+$) of SDF-1$^+$F4/80$^+$ cells, which was inhibited by AMD3100 (Fig. 6h).

Next, we explored the responses of other immune cells in WT and EC-p53KO tumours after radiotherapy. Immunofluorescence data showed that at 7 dpi, the population of granzyme B (GZMB)$^+$CD8$^+$ cytotoxic T cells was increased in EC-p53KO, but not WT tumours, compared to non-irradiated tumours. The GZMB$^+$CD8$^+$ population was significantly increased in irradiated EC-p53KO tumours (Supplementary Fig. 14a). The population of CD4$^+$Foxp3$^+$ regulatory T cells increased in WT tumours after radiation, and no difference was found between irradiated WT and EC-p53KO tumours (Supplementary Fig. 14b). Additionally, at 7 dpi, the MHCII$^+$ antigen-presenting cell population was significantly increased in EC-p53KO, but not WT tumours, compared to non-irradiated tumours (Supplementary Fig. 14c).

**EndMT-related vasculature in irradiated human lung cancer.** To evaluate the clinical relevance of our findings, we investigated whether tumour EndMT-related vasculature appears with populations of CD44v6$^+$ CSCs and M2 macrophages in lung tissues of patients with lung cancer who received radiotherapy (Supplementary Table 1). Because lung cancer patients who undergo neoadjuvant concurrent chemoradiotherapy followed by surgical resection are rare, we obtained human lung tissues after radiotherapy ($n = 10$), which were compared to non-irradiated human lung tissues ($n = 10$) (Supplementary Table 1). No comparable factors were found among other clinicopathologic characteristics (Supplementary Table 2). The patients who received radiotherapy received fractioned doses, not a single high dose, and our mouse studies (Supplementary Fig. 9) had shown significant EndMT after fractioned radiotherapy.

Tumour SMA$^+$CD31$^+$ vessels and coverage with SMA$^+$NG2$^+$ pericytes were correlated in >60% of tumour vessels in patients who underwent surgery following neoadjuvant or combined chemo-radiotherapy, compared to patients who did not receive radiotherapy (Fig. 7a). In non-irradiated cancer tissues, 38% of

tumour cells were CD44v6$^+$ (Fig. 7b). In contrast, 80% of residual tumour cells were CD44v6$^+$ in patients who underwent surgery following neoadjuvant radiotherapy (Fig. 7b). Coincidently, in irradiated cancer tissues, 62% of CD44v6$^+$ CSCs were OPN$^+$, whereas 18% OPN$^+$CD44v6$^+$ CSCs were detected in non-irradiated tissues (Fig. 7c). Additionally, in irradiated but not in non-irradiated tissues, SDF-1$^+$CD206$^+$CD68$^+$ M2 macrophages were predominantly detected. SDF-1$^-$iNOS$^+$CD68$^+$ M1 macrophages were hardly detected (<10%), although they were detected more in irradiated than in non-irradiated tissues (Fig. 7d, Supplementary Fig. 15). These data indicate that SDF-1$^+$ M2 macrophage polarization might occur in irradiated human tumours. Together, the occurrence of EndMT and subsequent phenomena in human tissues support the clinical relevance of our data.

**Discussion**

The findings of this study support a model for TRP53-regulated radiation-induced EndMT and tumour vasculature (Fig. 7e). EndMT induced abnormal recruitment of pericytes, including SMA$^+$, SMA$^+$NG2$^+$, and SMA$^+$desmin$^+$ cells, resulting in abnormal vasculature after radiotherapy. Moreover, EndMT ECs formed more tubules than non-EndMT ECs in vitro. While mature tumour vessels can show substantial pericyte coverage and abnormalities[37], non-irradiated tumour vessels show less pericyte coverage than EndMT-induced radioresistant vessels. We suggest that radiation-specific vasculature development may stem from tumour EndMT occurring shortly after radiotherapy, as pericyte–EC communication is regulated by direct physical contact and paracrine signalling[21]. Moreover, EndMT may give rise to pericyte-like cells within tumours[20] and be accompanied by pericyte coverage[29]. Recent findings suggest that tumours refractory to anti-vascular endothelial growth factor (anti-VEGF) therapy or chemotherapy feature perivascular cells with significantly increased SMA expression[20,38]. Thus, radiotherapy with anti-VEGF or anti-angiogenic therapy may be more efficient when combined with a strategy for inhibiting tumour EndMT.

Myeloid-lineage BMDCs are important in recurring irradiated tumours[7,8]. In our study, during early tumour regrowth, BMDCs recruited to EndMT vasculature after irradiation colocalized with SMA$^+$ cells, but not with ECs. We hypothesize that the recruited BMDCs support SMA$^+$ vessel maturation, as the origin of tumour-vascular pericytes was traced to mesenchymal progenitor cells or BMDCs[21,39]. A single, high dose (20 Gy) of non-curative radiation used to examine vasculature development during the initial-response, early-regrowth, and late-regrowth tumour phases was insufficient to eradicate all intratumour and peripheral tumour vessels or angiogenic vessels surrounding the tumours. In

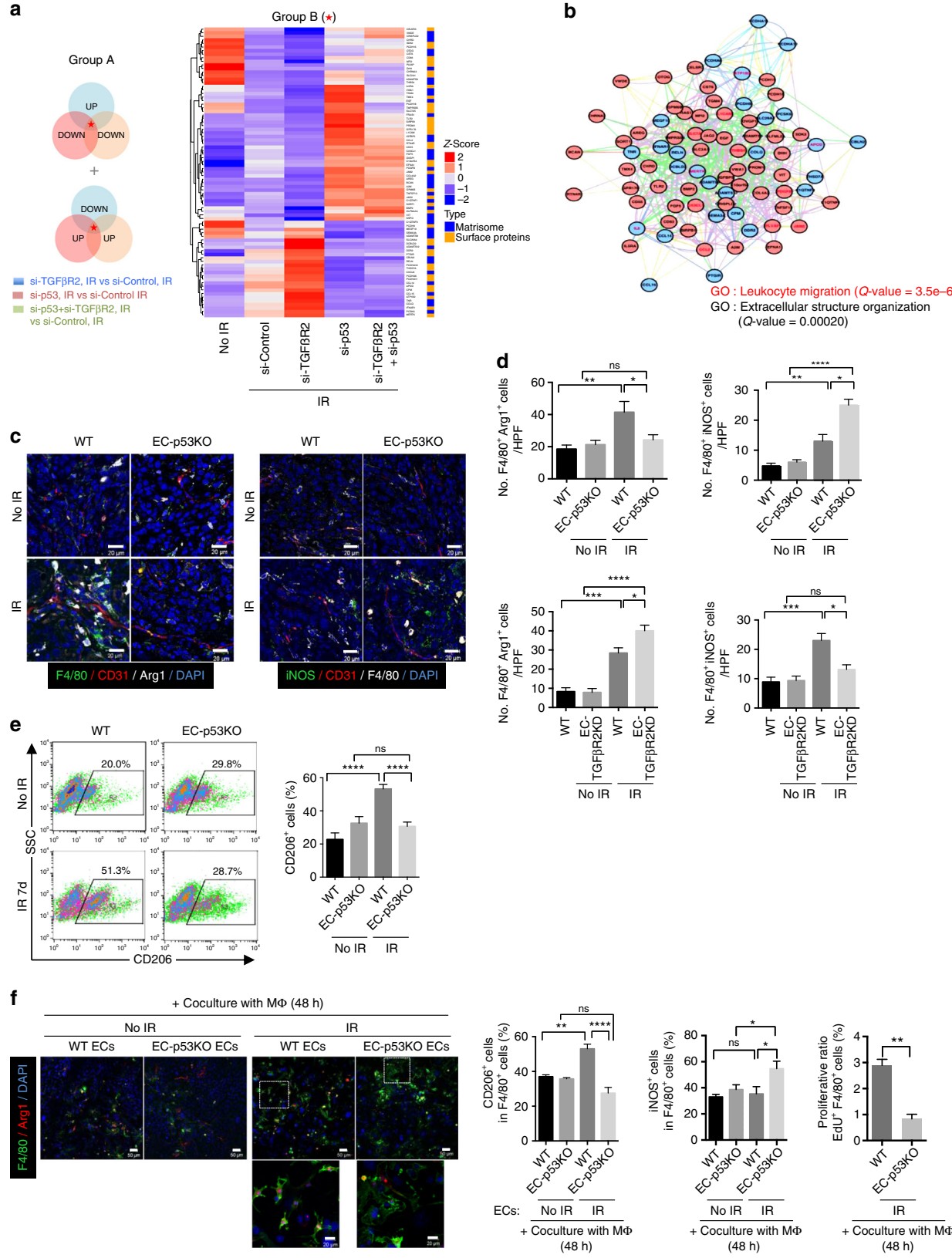

contrast, tumour EndMT was rarely detected in recurrent KP tumours after radiotherapy with 2 times 25 Gy (Supplementary Fig. 6). Therefore, angiogenesis may involve radioresistant tumour ECs and normal tissue ECs surrounding the tumour, which were unavoidably also irradiated during post-radiotherapy

tumour regrowth. Indeed, irradiated lung-tissue vessels show significant EndMT during radiation-induced lung fibrosis[25]. Thus, we suggest that modulating radiation-induced EndMT in both normal and tumour tissues is a viable strategy for increasing radiotherapy efficacy.

**Fig. 5** TRP53-regulated EndMT modulates the M1 and M2 populations of increased TAMs after radiotherapy. **a** RNA-seq data analysis of the samples represented in Fig. 4a. Venn diagram depicting differentially expressed genes in irradiated HUVECs. Reverse-regulated genes (>1.2-fold) between TGFβR2 knockdown+IR and *Trp53*-knockdown (with and without TGFβR2 knockdown)+IR conditions (compared with IR alone) were selected (subgroup A). Only matrisome- and surface-associated genes in subset A are shown in the heatmap generated by k-means clustering (subgroup B). **b** GeneMANIA network analysis of subset B showing significant enrichment for the GO term 'leukocyte migration'. Red-filled circles represent genes upregulated in the *Trp53* siRNA+IR group vs. the IR-alone group. Blue-filled circles represent genes upregulated in the *TGFβR2* siRNA+IR group vs. the IR-alone group. The names of leukocyte migration-related genes are indicated in red. **c** Immunofluorescence detection of F4/80, CD31, and Arg1, or iNOS, CD31, and F4/80 in KP tumours from WT and EC-p53KO mice, with or without irradiation (23 days after irradiation). Scale bar = 20 μm. **d** Quantification of F4/80⁺Arg1⁺ and F4/80⁺iNOS⁺ cells/field (magnification, ×200) using immunofluorescence images of KP tumours from WT, EC-p53KO, and TGFβR2KD mice, with or without irradiation (23 days after irradiation). **e** Ratio of CD206⁺ in F4/80⁺ cells from WT and EC-p53KO tumours 7 days after irradiation, as determined by flow cytometry. **f** Immunofluorescence staining of F4/80 and Arg1 in WT bone marrow-derived macrophages after coculture with KP tumour-derived ECs from WT and EC-p53KO mice for 48 h after irradiation. Scale bar = 50 μm (crop, 20 μm). Flow-cytometric analysis of CD206⁺, iNOS⁺, and EdU⁺ cells among F4/80⁺ cells (magnification, ×200) is shown. For (**d**–**f**), error bars indicate SEM; *p < 0.05, **p < 0.01, ***p < 0.001, ****p < 0.0001, ns not significant (one-way ANOVA for multiple comparison). The mice were treated with a single 20 Gy dose of irradiation. Data are representative of three independent experiments

TRP53 is regarded a key regulator of radiation responses and controlling cell survival and death; however, a reassessment of targeting TRP53 as a paradigm for cancer therapy is in order[40,41]. Our data revealed a novel role for TRP53 in the mesenchymal transition of the tumour microenvironment, especially tumour ECs, after radiotherapy.

TGFβ-related signalling potentially is also a key regulator of EndMT. Unexpectedly, EC-*Tgfbr2* knockdown enhanced radiation-induced EndMT, with strongly increased SMAD2/3 phosphorylation, suggesting that TGFβR2 deletion may compensatorily augment SMAD2/3 signalling via TGFβR1. Despite the essential effects of the TGFβ pathway on cancer progression, inhibiting TGFβ signalling in specific micro-environmental niches has produced conflicting results[42]. For example, suppressing TGFβ signalling in fibroblasts promoted tumour progression[43,44], whereas stromal TGFβR2 expression decreased as tumours progressed towards invasiveness[45].

We found that EC-p53KO tumours regulated OPN, an angiocrine factor acting on radioresistant CSCs, and EC receptors, which in turn controlled TAMs, in the radioresistant tumour microenvironment, supporting the observed role of tumour angiocrine factors in the crosstalk between CSCs and tumour vessels[46,47]. We suggest that OPN regulates tumour responses to radiotherapy. OPN secreted during tumour EndMT strongly accumulated on CD44v6⁺ CSC surfaces, implying that it acts as a tumour angiocrine factor towards CD44v6⁺ cells. Moreover, in vitro, non-proliferating CD44v6⁺ cells under severe hypoxia after irradiation became activated after OPN treatment. Indeed, dormant CD44v6⁺ CSCs were mainly detected in EC-p53KO tumours showing severely hypoxic regions after radiotherapy and CT26 cells prominently increased CD44v6 expression upon irradiation. Although activated CSCs are considered important in tumour relapse and escape, most CSCs are quiescent after radio- and chemotherapy[15,48]. Moreover, it remains unknown how awakened CSCs contribute to tumour relapse and escape. Based on our observations, we cautiously hypothesize that dormant (CD44v6⁺) CSCs may become awakened by angiocrine factors (such as OPN) secreted during tumour EndMT-related revascularization around hypoxic regions. Furthermore, we consider CD44v6⁺ CSCs more important than other CSCs in radiotherapy, as CD44v6⁺ populations were induced by radiation and might be mainly responsible for post-radiotherapy tumour regrowth and metastasis.

Irradiation seems to modulate tumour immunogenicity in conventional radiotherapy and stereotactic ablative radiotherapy[3]. High-dose radiation induces antigen production via tumour cell death, tumour-specific immunity, and tumour-vascular damage[49]. Radiation can attract immunosuppressive cells to the tumour microenvironment[17]. TAM modulation is considered important in cancer immunotherapy because of the tumour-promoting effects of enhancing an immunosuppressive microenvironment and tumour angiogenesis[50,51]. Klug et al.[52] reported that low-dose irradiation stimulates immunostimulatory macrophages, enabling cytotoxic T cells to infiltrate tumours, thus boosting tumour immunity. In the current study, we discovered that tumour vascularization via EndMT enhanced TAM polarization toward M2-like macrophages and proliferation after radiotherapy. M0 macrophages differentiated from BMDMs prominently showed Arg1⁺ M2 polarization in coculture with radiation-EndMT-derived tumour ECs. However, BMDMs mainly polarized to the iNOS⁺ M1 subtype during coculture with ECs from EC-p53KO mice, in which radiation-induced EndMT was inhibited. Accordingly, in vivo, at 1 dpi, tumours recruited monocytes and F4/80⁺ macrophages. At 7 dpi, M2 macrophages were mainly seen in WT tumours, while M1 macrophages were predominantly observed in irradiated EC-p53KO tumours, indicating that the tumour-vascular niche regulates TAM polarization after radiotherapy. Detailed effects of radiation-induced EndMT on other immune cells, including cytotoxic T and antigen-presenting cells, remain to be explored.

High SDF-1/CXCL12 expression in tumours aids in the capture of CXCR4-expressing monocytes[36,50]. Interestingly, strongly SDF-1⁺ monocytes and subsequently SDF-1⁺CD206⁺ M2 macrophages were recruited to KP tumours after radiotherapy. Furthermore, EndMT tumour-derived ECs highly expressing CXCR4 induced M2 polarization and proliferation of SDF-1⁺ macrophages, which was inhibited by *Trp53* silencing. AMD3100 significantly inhibited M2 SDF-1⁺ macrophage polarization in coculture with irradiated tumour ECs. Relevant to our findings with SDF-1⁺ macrophages, SDF-1 produced in monocytes contributes to macrophage differentiation[36]. In accordance with previous findings[53], SDF-1⁺ tumour cells were observed after radiation, but in this study, SDF-1 expression was higher in monocytes than in tumour stromal cells; this discrepancy may reflect a requirement for SDF-1 cleavage for cellular activity, as monocytic SDF-1 was not strongly detected with antibodies targeting the C terminus of SDF-1. We are currently studying SDF-1 cleavage in relation to monocyte differentiation.

The effect of radiation-induced tumour-vascular damage on TAM polarization, which was increased by EndMT, suggests that high-dose radiation (~10 Gy) can evoke stronger tumour immune responses than low-dose radiation, because it indirectly causes tumour cell death via vascular damage[11]. In this study, targeting EndMT more efficiently inhibited tumour regrowth after a single high dose of radiation than after fractionated radiotherapy. In fractionated radiotherapy, tumour EndMT was significantly

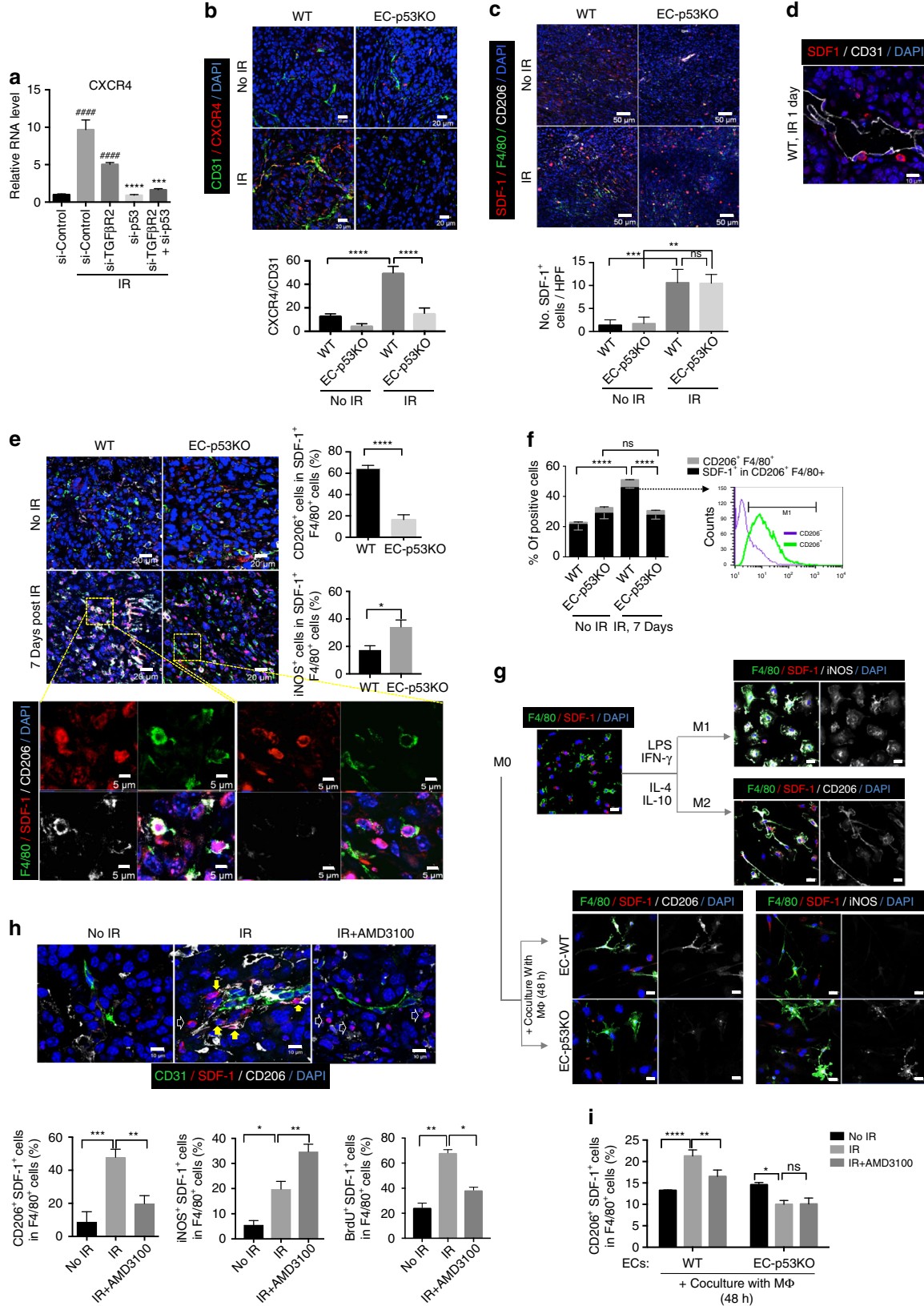

inhibited by EC-p53KO; however, CD44v6[+] CSC proliferation and tumour growth were not inhibited. Nevertheless, we cautiously suggest that in fractionated radiotherapy, targeting EndMT may enhance therapeutic efficacy when combined with inhibition of CSC proliferation.

In accordance with our findings in vivo, tumour EndMT was prominently seen in lung cancer tissues of patients who received irradiation, with large populations of OPN[+]CD44v6[+] CSCs and SDF-1[+]CD206[+] macrophages, even though the patients had received fractioned radiotherapy, indicating that

**Fig. 6** EC-*Trp53* deletion attenuates endothelial CXCR4 expression and M2 polarization of SDF-1$^+$ TAMs after a single 20-Gy irradiation. **a** RT-qPCR analysis of CXCR4 in HUVECs 48 h post 10 Gy irradiation. HUVECs were transfected with *TGFβR2*, *Trp53*, or *TGFβR2+Trp53* siRNAs 2 days before irradiation. si-TGFβR2 + IR or si-Control + IR versus si-Control group, ####$p < 0.0001$.si-p53 + IR or si-TGFβR2+si-p53 + IR versus si-Control + IR or si-TGFβR2 + IR group, respectively, ***$p < 0.001$ and ****$p < 0.0001$. **b** Immunofluorescence of CD31 and CXCR4 in KP tumours. CXCR4$^+$CD31$^+$ cell quantitation (magnification, ×200) is shown. Scale bar = 20 µm. **c** Immunofluorescence of SDF-1, F4/80, and CD206 in KP tumours, with or without irradiation (23 dpi). Quantification of SDF-1$^+$ cells (magnification, ×200) is shown. Scale bar = 50 µm. **d** Immunofluorescence of SDF-1 and CD31 in KP tumours from WT mice 1 dpi. Scale bar = 10 µm **e** Immunofluorescence of F4/80, SDF-1, and CD206 in KP tumours 7 dpi. Scale bar = 20 µm (crop, 5 µm). Quantification of CD206$^+$ and iNOS$^+$ cells among SDF-1$^+$F4/80$^+$ cells (magnification, ×200). **f** Flow-cytometric analysis of SDF-1$^+$ cells in CD206$^+$F4/80$^+$ cells of KP tumours at 7 dpi. **g** Immunofluorescence of F4/80, SDF-1, and CD206 in BMDMs after a 48-h coculture with KP tumour-derived ECs. As controls, BMDMs from WT mice were polarized to M1 or M2 macrophages by treatment with LPS (100 ng/ml) plus IFNγ (20 ng/ml), or IL-10 (20 ng/ml) plus IL-4 (20 ng/ml), respectively, for 48 h. Scale bar = 20 µm. **h** Immunofluorescence of CD31, SDF-1, and CD206 in KP tumours from WT mice at 7 dpi, with or without AMD3100 treatment. BrdU was used for in vivo proliferation assays. CD206$^+$SDF-1$^+$ TAMs, iNOS$^+$SDF-1$^+$ TAMs, and non-differentiated SDF-1$^+$ monocytes are marked with yellow, green, and open arrows, respectively. Quantification of CD206$^+$SDF-1$^+$ and iNOS$^+$SDF-1$^+$ cells among F4/80$^+$ cells (magnification, ×200) and flow-cytometric analysis of BrdU$^+$SDF-1$^+$ cells in F4/80$^+$ cells. Scale bar = 10 µm. **i** Flow-cytometric analysis of SDF-1$^+$CD206$^+$ cells in F4/80$^+$ cells of WT BMDMs after coculture with KP tumour-derived ECs from WT and EC-p53KO mice for 48 hpi, with or without AMD3100 treatment. Error bars indicate SD (**a**) or SEM (**c**, **e**, **f**, **h**, **i**); *$p < 0.05$, **$p < 0.01$, ***$p < 0.001$, ****$p < 0.0001$, ns not significant (one-way ANOVA). Data are representative of three independent experiments

radiation-induced CD44v6$^+$ tumour cells and SDF-1$^+$ M2 macrophages might be targeted for overcoming tumour radioresistance.

In conclusion, this study provides evidence that targeting radiation-induced EndMT is a new, promising strategy for enhancing tumour radiosensitivity by inhibiting radiation-induced abnormal tumour vasculature. Specifically, targeting vascular EndMT may be effective in regulating radioresistant CSC proliferation and maximizing the anti-tumour immunity of radiotherapy.

## Methods
**Mice**. All animal experiments were approved by the Institutional Animal Care and Use Committee of the Korea Institute of Radiological & Medical Sciences and are reported in accordance with the ARRIVE (Animal Research: Reporting of In Vivo Experiments) guidelines[54]. Specific pathogen-free C57BL/6 *Tie2*-Cre, *Trp53*$^{flox/flox}$, *Tgfbr2*$^{flox/flox}$, and LSL-*Kras*$^{G12D}$ mice were purchased from the Jackson Laboratory. *Tie2*-GFP mice were a kind gift from Dr. Gou Young Koh (Korea Advanced Institute of Science and Technology). Male *Tie2*-Cre mice were crossed with female *Trp53*$^{flox/flox}$, *Tgfbr2*$^{flox/flox}$, or *Trp53*$^{flox/flox}$*Tgfbr2*$^{flox/flox}$ (generated by crossing *Trp53*$^{flox/flox}$ and *Tgfbr2*$^{flox/flox}$) mice to generate *Tie2*-Cre;*Trp53*$^{flox/+}$, *Tie2*-Cre;*Trp53*$^{flox/flox}$, *Tie2*-Cre;*Tgfbr2*$^{flox/+}$, *Tie2*-Cre;*Trp53*$^{flox/+}$*Tgfbr2*$^{flox/+}$, or *Tie2*-Cre;*Trp53*$^{flox/flox}$*Tgfbr2*$^{flox/+}$ mice. *Tie2*-Cre;*Trp53*$^{+/+}$, *Tie2*-Cre;*Tgfbr2*$^{+/+}$, *Tie2*-Cre;*Trp53*$^{+/+}$*Tgfbr2*$^{+/+}$, or *Tie2*-Cre$^{-/-}$ littermates were used as controls. All animal experimental data shown are representative of three independent experiments. All experiments were conducted with 6–8-week-old mice. The mice had access to a standard diet and water ad libitum. All mice were anaesthetized with a combination of anaesthetics before being killed. For in vivo proliferation assays, mice received an intraperitoneal injection of 5-bromo-2-deoxyuridine (BrdU) (2 mg/mouse) (Sigma-Aldrich) 4 h before tumour harvesting. BrdU$^+$ cells were detected using an anti-BrdU mouse monoclonal antibody (1:500, Sigma-Aldrich).

**Human tissue specimens**. The analysis of lung cancer patient tissues was approved by the ethics committee of Severance Hospital, Yonsei University (Korea). In addition, a lung cancer tissue with information related to radiotherapy was purchased from Origene (cat. no. CT565899). Clinicopathological characteristics of 10 patients (Severance Hospital patient tissues, $n = 7$; Origene tissues, $n = 3$) who received radiotherapy and 10 patients (Origene tissues, $n = 10$) who did not receive radiotherapy are shown in Supplementary Table 1.

**Bone marrow transplantation**. CT26 cells ($5 \times 10^5$) were injected subcutaneously into the right thigh of 8-week-old male BALB/c mice. When the tumour volume reached 150–200 mm$^3$, tumours or whole mice were irradiated with a sublethal dose (8 Gy). Bone marrow cells were harvested from the tibias and femurs of male BALB/c donor mice and were fluorescently labelled with CM-Dil dye (1: 500, Molecular Probes), per the manufacturer's instructions. CM-Dil-labelled cells ($2 \times 10^7$) were suspended in 200 µl of Opti-MEM (Gibco) and intravenously injected into whole body-irradiated mice at 6 hpi. Tumours from control and transplanted mice were harvested at 7 dpi. KP tumour-bearing recipient mice were sub-lethally irradiated (9 Gy), and 9 h after irradiation, they received *Tie2*-GFP mouse-derived bone marrow cells ($1 \times 10^7$ cells) intravenously and were allowed to recover for 3 weeks.

**Histology and immunohistochemistry**. Tumour tissues were covered with optimal cutting temperature (OCT) compound (Sakura Finetek; VWR, IL, USA) and snap-frozen on dry ice or fixed in 10% (v/v) neutral-buffered formalin, embedded in paraffin, and sectioned. The sections were deparaffinized and stained as previously described[25]. The staining procedure for OCT sections was the same as that for paraffin-embedded sections, except that antigen removal was omitted. At least five images per section were acquired for quantification, and positively stained areas were evaluated with ImageJ software (http://imagej.net/).

**Tumour EC and macrophage coculture**. For coculture experiments, tumour ECs were plated in 6-well plates, with or without radiation. BMDMs ($1.2 \times 10^5$), differentiated by treatment with murine M-CSF every other day for 7 days, were placed on the tumour ECs. The cocultures were harvested at 48 h. Tumour ECs and BMDMs cultured alone were used as controls.

**AMD3100 treatment**. KP tumour cells ($2 \times 10^5$) were injected subcutaneously into the right thighs of C57BL/6 mice. When the tumours reached 150–200 mm$^3$, a single dose of radiation (20 Gy) was delivered using the X-RAD 320 platform (Precision X-ray). AMD3100 (Sigma-Aldrich) was dissolved in distilled water, diluted in phosphate-buffered saline, and administrated daily intraperitoneally (5 mg/kg) for 7 days.

**Syngeneic tumour models**. CT26 cells ($5 \times 10^5$) were injected subcutaneously into the right thighs of male BALB/c mice. KP tumour cells ($2 \times 10^5$) were injected subcutaneously into the right thighs of *Tie2*-Cre;*Trp53*$^{flox/+}$, *Tie2*-Cre;*Trp53*$^{flox/flox}$, *Tie2*-Cre;*Tgfbr2*$^{flox/+}$, *Tie2*-Cre;*Trp53*$^{flox/+}$*Tgfbr2*$^{flox/+}$, *Tie2*-Cre;*Trp53*$^{flox/flox}$*Tgfbr2*$^{flox/+}$, *Tie2*-Cre;R26R$^{tdTomato}$;*Trp53*$^{flox/flox}$, and VE-cadherin-CreERT2;*Trp53*$^{flox/flox}$ mice and control littermates. Tumour volumes were determined according to the formula ($L \times W \times H$)/2, using the calliper-measured tumour length ($L$), width ($W$), and height ($H$). When tumour volumes reached 150–200 mm$^3$, radiation was delivered, using the X-RAD 320 platform (Precision X-ray). For OPN blocking experiments, mice were intraperitoneally injected with an anti-mouse OPN neutralizing antibody (20 µg/injection; R&D systems) four times, every 2 days.

**Mouse experiments**. To visualize functional vasculature, mice were intravenously injected with Hoechst 33342 (20 mg/kg; Sigma) 1 min prior to killing. For hypoxia assessment, pimonidazole (60 mg/kg) was administered intraperitoneally. After 45 min, the mice were killed and tumours were harvested, fixed, and immunostained using the Hypoxyprobe Plus Kit (Hypoxyprobe) per the manufacturer's instructions. To evaluate tumour metastasis, metastatic nodules on lung surfaces observed by the naked eye were counted. To visualize tumour vessel leakage, mice were intravenously injected with fluorescein isothiocyanate–dextran (70 kDa, 40 mg/kg; Sigma) 5 min before killing.

**Flow-cytometric analysis**. At day 7, tumour samples were harvested and digested into single-cell suspensions. Cells were stained with macrophage-specific antibodies. The cells were fixed and permeabilized using the Cytofix/Cytoperm Kit (BD Biosciences) and stained with an anti-SDF1 antibody. Then, the cells were analysed on a FACSCalibur flow cytometer (Becton Dickinson, San Jose, CA). Gating strategies for flow cytometry analysis are shown in Supplementary Figure 16.

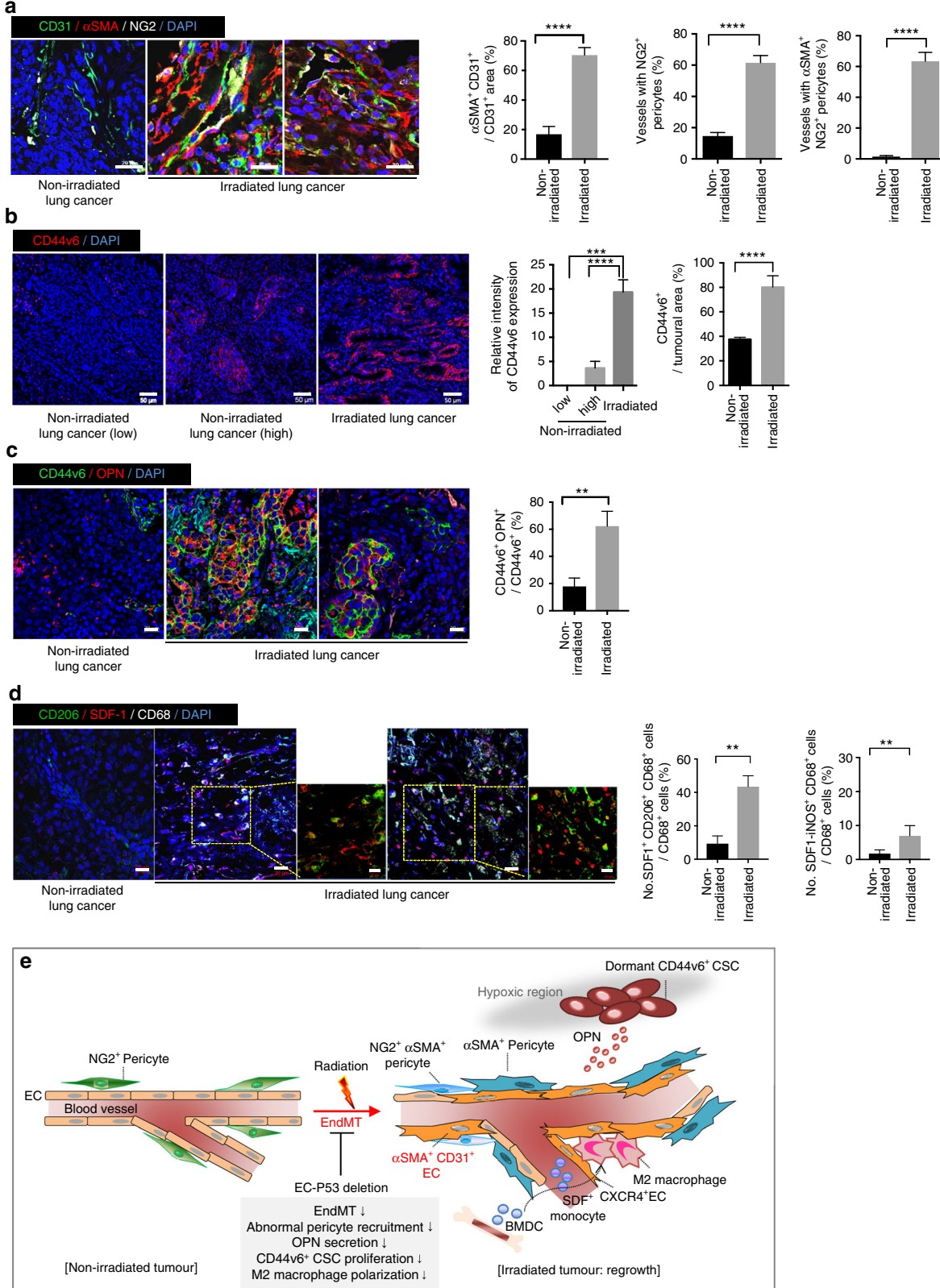

**Flow-cytometric analysis of 5-ethynyl-2′-deoxyuridine (EdU) incorporation**. For in vitro proliferation assays, KP cells were irradiated with gamma rays from a [137Cs] source (Atomic Energy of Canada) at 3.81 Gy/min. To establish hypoxia, cells were incubated in a Forma 1025/1029 Anaerobic Chamber (Thermo Fisher Scientific) flushed with 0.5% or 1% $O_2$ for severe or mild hypoxia, respectively. Cells were incubated with 10 μM EdU for 2 h, trypsinized, and surface-stained with an anti-CD44v6 antibody (Novus Biologicals, USA).

The cells were labelled using the Click-iT Alexa Fluor 488 Assay Kit (C10420; Thermo Fisher Scientific) and analysed on a FACSCalibur (Becton Dickinson, San Jose, CA).

**Generation of BMDMs**. Femurs and tibiae were collected from adult male C57BL/6 mice, and the bones were flushed using syringes filled with RPMI-1640 medium

**Fig. 7** EndMT-related phenomena in human lung cancer tissues from patients treated with or without radiotherapy. Sections from non-irradiated ($n = 10$) and irradiated ($n = 10$) lung cancer tissues were analysed (a–d). **a** Immunofluorescence detection of CD31, αSMA, and NG2 in irradiated tissues. Quantification of αSMA⁺CD31⁺ among the total CD31⁺ cells or vessels with αSMA⁺NG2+ pericytes (magnification, ×100). Scale bar = 20 µm. **b** Immunofluorescence detection of CD44v6 expression in non-irradiated and irradiated tissues. Quantification of the CD44v6 intensity and CD44v6⁺ area/field (magnification, ×100) is shown. Scale bar = 50 µm. **c** Immunofluorescence detection of CD44v6 and OPN expression in irradiated tissues. Quantification of CD44v6⁺OPN⁺ cells in the total CD44v6⁺ area (magnification, ×100) is shown. Scale bar = 20 µm. **d** Immunofluorescence detection of CD206, SDF-1, and CD68 in irradiated human lung cancer tissues. Scale bar = 20 µm (crop, 10 µm). Quantification of SDF-1⁺CD206⁺CD68⁺ cells in SDF-1⁻iNOS⁺CD68⁺ cells total CD68⁺ cells (magnification, ×100) is shown. **a–d** Error bars indicate SEM; **$p < 0.01$, ***$p < 0.001$, ****$p < 0.0001$ (Student's *t*-test). **e** Model for TRP53-regulated irradiation-induced EndMT and tumour vasculature. Radiation-induced tumour EndMT causes aggressive tumour-vasculature growth, recruiting abnormal αSMA⁺ or NG2⁺αSMA⁺ pericytes. Firstly, vascular ECs occur in EndMT secreted OPN, which trigger the proliferation of radioresistant dormant hypoxic CD44v6⁺ CSCs with metastatic potential. Secondly, EndMT cells overexpressing CXCR4 serve as a reservoir for M2 macrophages polarized from SDF-1-positive monocytic cells. These phenomena synergistically affect aggressive tumour regrowth after radiotherapy. However, endothelial TRP53 deletion inhibits radiation-reduced EndMT and aberrant tumour vasculature. Subsequently, inhibited OPN secretion and CXCR4 expression can synergistically inhibit tumour regrowth after radiotherapy. Our findings suggest that targeting radiation-tumour EndMT, as the marker of radioresistant tumour, may enhance radiotherapy efficacy both by inhibiting the reactivation of dormant CSCs and by promoting radiation-antitumour immune responses, representing a potentially viable, new therapy

(WELGENE, Gyeongsan, South Korea) supplemented with 1% penicillin/strepto-mycin and 10% foetal bovine serum (FBS). Red blood cells were lysed using lysis buffer (catalogue number R7757; Sigma). Bone marrow cells were seeded onto 100 mm culture dishes and incubated at 37 °C under 5% CO₂. Non-adherent cells were removed at day 3. The attached cells were cultured with fresh medium containing murine M-CSF every other day for 7 days. On day 7, M1 macrophages were characterized by the expression of inducible nitric oxide synthase (iNOS) and induced with LPS (100 ng/ml) or IFN-γ (20 ng/ml), whereas M2 macrophages were characterized by the expression of CD206 and induced with IL-4 (20 ng/ml) or IL-10 (20 ng/ml).

**Immunoblotting and immunohistochemistry.** Staining for immunohistochem-istry, immunofluorescence, and flow cytometry was carried out using primary antibodies against CD31 (Immunofluorescence 1:200; R&D Systems; #AF3628, Immunoblotting 1:1000; Santa Cruz; #sc-1506) and αSMA (1:1000; Sigma-Aldrich; #A5228); vimentin (1:10,000; Santa Cruz; #sc-6260), phospho-SMAD2/3 (1:1000, Cell Signaling; #8828), phospho-SMAD1/5 (1:1000, Cell Signaling; #9516), p53 (1:1000; Santa Cruz; #sc-126), CD44 (1:200; Santa Cruz; #sc-7297), GFP (1:100; Santa Cruz; #sc-9996), SMAD2/3 (1:1000, Cell Signaling; #3102), TGFβR1 (1:1000; Santa Cruz; #sc-339), TGFβR2 (1:1000; Santa Cruz; #sc-17792), OPN (1:100; Santa Cruz; #sc-21742), iNOS (1:100; Santa Cruz; #sc-21860), Nanog (1:200; Santa Cruz; #sc-134218), Oct4 (1:200; Santa Cruz; #sc-5279), Sox2 (1:200; Santa Cruz; #sc-20088), and CD4 (1:200; Santa Cruz; #sc-52385); EpCAM (1:200; Biorbyt; #orb10618); CD44v6 (1:500; Novus Biologicals; #NEB100-64818) and arginase 1 (1:2000; Novus Biologicals; #NBP1-32731); NG2 (1:500; Millipore; #AB5320); Ki67 (1:200; Acris; #DRM004); RasG12D (1:100; New East Biosciences; #26036); β-Catenin (1:200; BD; #610153); desmin (1:200; Abcam; #ab8592), ALDH1 (1:200; Abcam; #ab52492), γH2AX (1:200; Abcam; #ab2893), F4/80 (1:200; Abcam; #ab6640), CD206 (1:200; Abcam; #ab8918), SDF-1 (1:200; Abcam; #ab18919), MHCII ((1:200; Abcam; #ab25333), CD8 (1:200; Abcam; #ab22378), Granzyme B (1:200; Abcam; #ab4059), and Foxp3 (1:200; Abcam; #ab54501). Scanned images of western blots are shown in Supplementary Figure 17. Hoechst 33342 staining was applied directly to frozen OCT sections. Collagen deposition was assessed using Masson's trichrome stain (Sigma-Aldrich). For TUNEL (terminal deoxynucleotidyl transferase dUTP nick end labelling) staining, the DeadEnd Fluorometric TUNEL System (Promega) was used. To visualize actin stress fibres, cells were stained with Alexa Fluor 488-conjugated phalloidin (1:40; Invitrogen), which specifically binds polymerized F-actin. At least five images per section were acquired for quantifi-cation, and positively stained areas were evaluated with ImageJ software (http://imagej.net/).

**RNAscope in situ hybridization and immunofluorescence staining.** Frozen OCT sections were used for in situ hybridization assays. *Trp53* mRNA was stained using a *Trp53*-specific probe (Advanced Cell Diagnostics, #513001-C3; NM_011640.3, region 159–1238) and the RNAscope Fluorescent Multiplex Reagent Kit (Advanced Cell Diagnostics, #320850), according to the manu-facturer's instruction. Thereafter, sections were blocked with normal horse serum for 30 min at room temperature and incubated with a primary antibody overnight at 4 °C. *Trp53* mRNA transcripts and VE-cadherin were visualized with TSA Plus fluorescein (PerkinElmer) and an Alexa Fluor 647-labelled secondary antibody (Invitrogen), respectively.

**Cell culture and treatments.** HUVECs, human pulmonary microvascular endo-thelial cells (HPMECs), and human pericytes were obtained from PromoCell (Sungwoo Life Science, Uijeongbu, South Korea) and were cultured in Endothelial Cell Growth Medium 2, Endothelial Cell Growth Medium MV2, and Pericyte Growth Medium, respectively (Promocell) under 5% CO₂. HUVECs, HPMECs, and pericytes were used within nine passages. CT26 mouse colon carcinoma cells were a kind gift from Dr. Sam S. Yoon. The cells were cultivated in RPMI-1640 (WELGENE) supplemented with 10% FBS. CT26 cells stably expressing enhanced green fluorescent protein (EGFP) were established by transduction with a lentiviral vector expressing EGFP under the control of the cytomegalovirus promoter (Invitrogen), according to the manufacturer's protocol. KP cells were isolated from lung adenocarcinoma samples of LSL-KrasG12D;Trp53flox/flox mice[30]. Primary lung tumours were digested for 1 h at 37 °C with 1 mg/ml collagenase I (Gibco) in phosphate-buffered saline and cultured in Dulbecco's modified Eagle's medium (WELGENE) supplemented with 10% FBS. KP cells at passages 3–6 were used for transplantation. For silencing experiments, cells were transfected with siRNAs targeting *Trp53* and *Tgfbr2* as well as control siRNA (Santa Cruz Biotechnology) using Lipofectamine 2000 (Invitrogen) according to the manufacturer's recom-mendations. Cells were irradiated with gamma rays from a [¹³⁷Cs] source (Atomic Energy of Canada) at 3.81 Gy/min. For the analysis of secreted proteins, condi-tioned media were collected between days 6 and 7 of HPMEC culture after irra-diation and assayed using Proteome Profiler Human Soluble Receptor Array, Hematopoietic Kit (R&D Systems, ARY001), according to the manufacturer's instructions. For the analysis of cellular proteins, immunocytochemistry was per-formed as previously described[26]. OPN released by HUVECs into the culture medium was measured using a human OPN Quantikine ELISA Kit (R&D Sys-tems), following the manufacturer's instructions.

**RNA-seq analysis.** Total RNA was isolated from HUVECs, and RNA quality was assessed using an Agilent 2100 Bioanalyzer (Agilent Technologies). RNA-seq libraries were constructed using the SENSE mRNA-Seq Library Prep Kit (Lexogen), according to the manufacturer's instructions, and were sequenced as 100 bp paired-end runs on the HiSeq 2000 platform (Illumina). RNA-seq reads were mapped to UCSC hp19 using TopHat software[55]. Fragments-per-kilobase-of-transcript values were calculated with Cufflinks[55]. Genes with a maximal log2-transformed read count <5 across samples were excluded from the analysis. To identify differentially expressed genes, we compared the direction of fold changes (>1.2) against irra-diation (IR) alone. Reverse-regulated genes (>1.2-fold) between TGFBR2 knock-down+IR and p53 knockdown (with and without TGFBR2 knockdown)+IR compared with IR alone were selected. To analyse angiocrine factors and pheno-typic changes in endothelial cells, matrisome gene lists and surfactome gene lists were used. The matrisome comprises core matrisome-related proteins, including extracellular matrix (ECM) glycoproteins, collagens, and proteoglycans, as well as ECM-associated proteins, including ECM-affiliated proteins, ECM regulators, and secreted factors[56]. The surfactome represents proteins at the plasma membrane, including G protein-coupled receptors, receptor tyrosine kinases, and integrins[57]. Among the differentially expressed genes, matrisome and surfactome genes were displayed in R using the ComplexHeatmap package[58] and were used as input for the GeneMANIA 3.4.1 plugin[59] within Cytoscape 3.4.0[60] for functional enrichment analysis. RNA-seq data generated in this study are available on Gene Expression Omnibus (GEO, accession number GSE118538)

**RT-qPCR analysis.** RNA was isolated using TRI reagent (MRC, Cincinnati, OH, USA), and 1 µg of RNA was used to synthesize complementary DNA with an Omniscript RT Kit (Qiagen, Hilden, Germany). PCR was conducted in triplicate on the CFX96 TM Real-Time system (Bio-Rad, Hercules, CA, USA), using qPCR SYBR Green master mix (Invitrogen). For each sample, target gene expression was normalized against the geometric mean of the reference gene *GAPDH*. Primers are listed in Supplementary Table 3.

**Statistical analyses**. Student's $t$-test (Figs. 2a–c, 3b, lower, 3f, right, 3g, left, 4d, 5f, right, 6e, 7a–d) and analysis of one-way analysis of variance (ANOVA) for multiple comparison (for all others) in GraphPad Prism version 5.0 were used to compare experimental groups. A $p$ value < 0.05 was regarded significant. Experimenters were blinded to group assignments and outcome assessments.

## Data availability

The RNA-seq data were deposited in Gene Expression Omnibus (GEO) with accession number GSE118538. All other relevant data are available from the corresponding author upon reasonable request.

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

## Acknowledgements

This work was supported by grants from the National Research Foundation (NRF-2013M2A2A7043580, NRF-2017M2A2A7A02019482, and NRF-2017R1A2B2004156), as well as a grant from the Korea Institute of Radiological & Medical Sciences (KIRAMS, 50531–2018) funded by the Ministry of Science and ICT (MSIT), Republic of Korea.

## Author contributions

Y.-J.L. and S.-H.C. conceived and designed the study. Y.-J.L., S.-H.C., A.-R.K., and J.-K. N. performed the experiments with help from J.-M.K., J.-Y.K., H.S., H.-J.L. and J.C. Y.-J. L., S.-H.C., and A.-R.K. interpreted the data. Y.-J.L. wrote the paper.

## Additional information

**Competing interests:** The authors declare no competing interests.

