## [Peer Review File · Nature Communications]

Reviewers' Comments:

Reviewer #1:

Remarks to the Author:

In this manuscript Choi et al. assert that "radiation leads to tumour EndMT, which may induce abnormal SMA+NG2+ pericyte recruitment to irradiated WT tumour vessels, resulting in increased tumour growth after radiotherapy." The investigators present data from several in vitro studies of HUVECs and employ endothelial conditional knockout of Trp53 and TGF β R2 combined with syngeneic subcutaneous tumor implant (KP, lung cancer model, and CT26 cells) to show that p53 and TGFBR2 play opposing roles in modulating radiation-induced EndMT, the former as a driver and the latter as a repressor, and that conditional deletion of these genes promotes tumor regrowth after radiation. The rationale for evaluating these genes is not clearly explained in the manuscript, but the investigators show that radiation-induced increases in the α SMA+CD31+ cells are regulated by these genes and that radiation-induced EndMT is associated with accelerated tumor regrowth after radiation. The authors show that radiation-induced EndMT is associated with an increase CD44v6+ cancer stem cells that is suggested to be the result of increased osteopontin secretion and skewing the immune response toward an M2 phenotype. Finally, the authors provide preliminary data that suggests that EndMT may occur in humans treated with radiotherapy for lung cancer. Overall the manuscript sheds light on an important clinical problem. However, the manuscript lacks critical evidence to prove that EndMT drives the mechanisms that they have identified as mediators of post-radiation tumor regrowth

- 1) The rationale for targeting p53 and TGF β R2 for endothelial conditional knockout is unclear. The sole explanation given is "To this end, we generated EC-specific Trp53- or Tgfb2-deletion mice as these genes may regulate radiation-induced EndMT." It seems highly likely that knockout of these genes would affect many biological processes independent of any effect on EndMT. The authors should provide a better justification for their choice of models and provide evidence for the mechanism by which p53 and TGF β R2 signaling modulate EndMT itself (i.e. regulation of transcription factors known to be implicated in EndMT).
- 2) The investigators show that irradiation increases the number of CD31+ α SMA+ and NG2+ α SMA+ cells in a syngeneic tumor implant model, and they report that endothelial knockdown/knockout of p53 and TGF β R2 modulate this effect. They conclude that the changes in the absolute and relative numbers of these cell populations are due to EndMT, but no lineage tracing experiments were performed. A lineage tracing experiment is necessary to prove that EndMT contributes to changes in the tumor pericyte population.
- 3) Why are the percentages of α SMA+CD31+/CD31+ and NG2+ CD31+/CD31+ cells different in the WT group in Figure 1e and 1g (~70%) vs Figure 2d and 2e (~30%)? The control groups should have similar results. Please explain the discrepancy.
- 4) The central mechanistic role of osteopontin needs to be further proven. The investigators must show that osteopontin inhibition can rescue the phenotype of Tgfb2 conditional knockout mice.
- 5) In Fig 4g the investigators report that irradiation induces an increase in percentage of OPN+CD31+/CD31+ cells, which is blocked by p53KO. However, this percentage does not demonstrate that the population of cells that have undergone EndMT are the source of OPN. The investigators must demonstrate that irradiation induces changes in osteopontin expression occur in cells that have undergone EndMT.
- 6) Several markers of cancer stem cells have been proposed. The authors justify the use of CD44v6+ given its mechanistic relationship with OPN. However, the investigators must demonstrate that the CD44v6+ cells display other markers of stem-ness to make the claim that OPN induces proliferation of cancer stem cells.
- 7) The investigators report that "Tumour EndMT-recruited abnormal SMA+NG+ [NG2+] pericytes were detected in >60% of tumour vessels in tissues of patients who underwent surgery following neoadjuvant or combined chemo-radiotherapy, compared to patients who did not receive radiotherapy." First, this statement should be revised to simply report the findings without asserting that the observed findings are due to EndMT recruitment, as that cannot be proven using this approach. At best, their findings are correlative. Secondly, the data are presented as

aSMA+CD31+/CD31+ cells and NG2+CD31+/CD31+ cells, and it is unclear if 60% of the pericytes are double-positive (aSMA+ and NG2+). Quantification should be provided.

8) The authors report that human lung cancer samples that had been irradiated had a higher proportion of SD1+CD206+ (M2-type) macrophages. However, they do not quantify M1 type macrophages. The investigators should show relative percentages of M1/M2 macrophages to determine the degree of polarization.

9) The analysis of the pericytes recruitment as a consequence of the irradiation is not clear. The investigators refer to recruitment in terms of appearance of NG2+ cells in the tumor vasculature and they presented a quantification of the CD31+NG2+ double positive cells. This interpretation is misleading as pericytes are physically in contact with endothelial cells but do not express endothelial markers. Are the authors truly looking at the pericytes coverage of the tumor vasculature or is NG2 a mesenchymal marker expressed by the CD31+ endothelial cells as consequence of the activation of the EndMT program?

10) The transition from one mechanism to another, as it relates to the tumor biology the authors are focusing on is sometimes not clear. For example, was CD44v6 the only CSCs marker with a differential expression in the irradiated WT compared to the irradiated p53 KO tumors? The expression of EpCAM, CD133 and ALDH is presented only in the irradiated versus non-irradiated WT tumors, and no information is provided regarding their status in the p53 KO model. Therefore is not clear whether CD44v6 was chosen because among the CSCs marker it is the most upregulated in the irradiated WT tumors, or if because it is the only marker whose expression changes in the WT IR compared to p53 KO IR.

11) Is the increase in hypoxia observed in the p53 KO tumors caused by an increase in the vascular leakage due to the loss of pericytes coverage when EndMT is inhibited?

12) There is no data demonstrating that p53 was successfully knock-down/knock-out in the Tie2-Cre;p53 mouse model.

13) There are some discrepancies in the growth curves of KP WT tumors across the different mouse models (compare Fig. 1b, 2b and 2g). In particular, the growth rate of the WT tumor in Fig.2b seems to be considerably slower compared to the other, thus highly affecting the conclusion that tumors in the TGFbRII KD model growth more than the WT. Please clarify.

14) In Figure 4c/d, the authors supported the result regarding the OPN transcript levels in the HUVEC (4c) with a cytokine array performed on CM media from a completely different endothelial cells model (human pulmonary microvascular endothelial cells). What is the level of secreted OPN in the HUVEC under all the different conditions presented in Fig.4c?

15) Statistical analysis is missing in all tumor growth curves.

16) Please add quantification of the staining in the following figures: 6b, S3d, S4h.

17) Figure S1c: what are the CD31+ area and vessel diameter in the regression stage?

18) Regarding the data shown in Figure S4a-b, the authors stated that the levels of pSmad2/3 and EndMT after IR or TGFb1 treatment differentially changed depending on which TGFb receptor is knock-down. However only pSmad2/3 levels are shown in the figure, no data on the EndMT phenotype.

Reviewer #2:

Remarks to the Author:

NCOMMS-17-33270

Choi et al. "Tumor-vasculature development via the endothelial-to-mesenchymal transition after radiotherapy controls CD44v6+ cancer cell and macrophage polarization "

The authors study the effect of 20Gy of radiation in mouse preclinical models (one colon cancer and one lung cancer KRAS p53) of cancer on the associated tumor vasculature and then on a cancer stem cell subpopulation. Their major findings involve detection of endothelial cell mesenchymal transition after radiation and the effect this has on protecting a hypoxic cancer stem cell subpopulation. As part of this they use knockout models (TP53 and TGFRB2 receptor) to modulate effects. As part of this they study the presence of M2 (immune suppressive) and M1

(immune stimulatory) macrophages after radiation and the mechanisms involved (including osteopontin secretion). They end by providing data on immune histochemical analyses of human lung cancers for various endothelial cell, macrophage, and tumor stem cell markers with and without radiation treatment. They conclude: "Our findings suggest that targeting tumour EndMT might enhance radiotherapy efficacy by inhibiting the re-activation of dormant hypoxic CSCs and promoting anti-tumour immune responses. "

Comments to the authors:

The manuscript is reviewed in the context for urgent need to understand the effects of radiation on tumor treatment and potential immuno-stimulatory vs. immuno-inhibitory and cancer stem cell ablative vs. protective effect of radiation and potential ways to develop new therapies. This is particularly important given the recent advances of using stereotactic ablative radiation therapy for lung cancer including its role in local consolidation for oligometastatic disease.

All of the experiments are technically well done and presented in great detail. There are several issues the authors need to address. Of all of the issues, the most important is the information required to understand the studies of the human tissues.

1. The way the paper is presented, as a whole is very difficult to read to identify their key points. They should include some kind of summary schema of their current findings that would provide a road map for integrating their findings. As part of this they can indicate what they think the key biomarkers would be and potential therapeutic targets.

2. All of their animal experiments appear to be done with 20 Gy of radiation (a very high dose). However, the methods provide no indication of how this was delivered or the fraction(s) used. While this can easily be corrected in the Methods I found the omission of this basic piece of information to be very serious. Because of the use fractionated vs. stereotactic radiation therapy in the treatment of human lung cancer, it would be very important to know if the fractionation of the radiotherapy influences any of their key results. Thus, standard doses to patients are given in 1.5-2 Gy fractions vs. very high dose single fraction ablative radiation therapy. Is there a difference in the EndMT? Obviously we need to know this whatever the answer is.

3. Some of the most important data are those from patients in Figure 7 and Supplemental Table 1. However, in reading over the patient numbers and methods I have no idea which data are used in Figure 7 and how the overall two datasets were used for the studies. I cannot stress strongly enough to the authors that I was very impressed with their work, but when I dug into the details about the human studies I was very disappointed by the information they provided. While I hope they can provide this key information, its lack in the manuscript at this point was a very big negative for me. I suspect this is the Origene N = 13 patient dataset but, if so, what data were generated from the Severance Hospital data set. In addition, in the Methods they discuss 27 tumors of which 3 received radiation therapy. In addition, from Table S1 it is clear the patients received chemotherapy with the radiation therapy so which treatment is responsible is, of course, clouded. Now such samples are hard to come by, but it is important for the authors to address this limitation in the discussion. Also, we need to know the comparison demographics of the tumors that did and did not receive radiation (such as gender, histology, smoking status, stage, and if possible oncogenotype). Were the two groups comparable? Finally, the mouse studies were with 20 Gy while the patient studies were with fractionated radiation therapy. Thus, it is very important the authors provide preclinical data that can be matched with the treatment given to their patient specimens.

Reviewer #3:

Remarks to the Author:

Choi et al. present an interesting story on the effect of irradiation on p53-dependent endothelial to mesenchymal transition. the experiments are elegantly conducted and well controlled.

Several points in the manuscript are less well explored while others might need some clarification.

General:

The biggest concern from the reviewer is the lack of mechanisms shown in the manuscript. The authors show that deletion of p53 in endothelial cells results in increased tumor control reducing of cell death accompanied by reduced EndMT. EndMT in turn is inhibited via hypoxia (potentially due to reduced cell death) and M2 macrophages. Loss of p53 results in ablation of SDF1 and reduced macrophage recruitment. in a beautiful experiment in figure 5f the authors prove the M2 macrophages drive EndMT directly but not through which mechanisms. This should be a small and testable list of possibilities and thus should be explored to show mechanism and provide therapeutic opportunities.

the manuscript would benefit greatly from a summary schematic

Figure 1: when were the tumors irradiated?

Data showing decreased cell death in p53-KO mice should be included in the main figure.

Figure 2: d/e/i in i) the authors compare double KO with WT (60% and 40% SMA and NG2+ cells) while in d) WT is 40% and in e) Wt is 30%. This could be due to experimental difference but should this be the case the authors need to include single KO in figure i.

Figure 3: e is missing f exists twice.

Figure 5: The authors solely focus on macrophages but other immune cells could be affected likewise. Are T cell numbers different? other MHC II antigen presenting cells?

Responses to the Reviewers' comments

We thank the reviewers for the helpful comments and suggestions, which have greatly helped us in improving the manuscript. Please find our point-by-point responses below. Revisions in the manuscript are highlighted in blue.

Reviewer #1 (Remarks to the Author):

In this manuscript Choi et al. assert that “radiation leads to tumour EndMT, which may induce abnormal SMA+NG2+ pericyte recruitment to irradiated WT tumour vessels, resulting in increased tumour growth after radiotherapy.” The investigators present data from several in vitro studies of HUVECs and employ endothelial conditional knockout of Trp53 and TGFβR2 combined with syngeneic subcutaneous tumor implant (KP, lung cancer model, and CT26 cells) to show that p53 and TGFβR2 play opposing roles in modulating radiation-induced EndMT, the former as a driver and the latter as a repressor, and that conditional deletion of these genes promotes tumor regrowth after radiation. The rationale for evaluating these genes is not clearly explained in the manuscript, but the investigators show that radiation-induced increases in the αSMA+CD31+ cells are regulated by these genes and that radiation-induced EndMT is associated with accelerated tumor regrowth after radiation. The authors show that radiation-induced EndMT is associated with an increase CD44v6+ cancer stem cells that is suggested to be the result of increased osteopontin secretion and skewing the immune response toward an M2 phenotype. Finally, the authors provide preliminary data that suggests that EndMT may occur in humans treated with radiotherapy for lung cancer. Overall the manuscript sheds light on an important clinical problem. However, the manuscript lacks critical evidence to prove that EndMT drives the mechanisms that they have identified as mediators of post-radiation tumor regrowth

1) The rationale for targeting p53 and TGFβR2 for endothelial conditional knockout is unclear. The sole explanation given is “To this end, we generated EC-specific Trp53- or Tgfbr2-deletion mice as these genes may regulate radiation-induced EndMT.” It seems highly likely that knockout of these genes would affect many biological processes independent of any effect on EndMT. The authors should provide a better justification for their choice of models and provide evidence for the mechanism by which p53 and TGFβR2 signaling modulate EndMT itself (i.e. regulation of transcription factors known to be implicated in EndMT).

Response: We thank the reviewer for this insightful comment. We examined the effects of *Trp53*- and *Tgfbr2*-deleted HUVECs on radiation-induced EndMT-related transcriptional factors *in vitro*, the results of which are shown in Supplementary Fig. 1a, b.

The text in the Results section (p. 4) has been added as follows: ‘At 48 h post-irradiation, small-interfering RNA (siRNA)-mediated *Trp53* silencing in human umbilical vein ECs (HUVECs) markedly inhibited irradiation-induced mRNA levels of *Snail1*, *Snail2*, and *Zeb2*, which encode transcription factors implicated in EndMT, compared to control siRNA-treated cells, whereas *Tgfbr2* knockdown increased these levels (Supplementary Fig. 1a, b)’.

Additionally, we have provided a better justification for our study design using EC-specific *Trp53*- or *Tgfbr2*-deletion mice by revising the text and citing additional references.

Results (p. 4): To provide more supporting evidence for selecting the *Trp53* and *Tgfbr2* genes as regulators of radiation-induced EndMT, we have now presented the *in-vitro* findings in a first subsection of the Results.

Trp53 and Tgfbr2 conversely regulate EndMT *in vitro*. We previously reported radiation-induced EndMT in several EC types^{24, 25, 26}. TRP53 is considered a key regulator of radiation responses in endothelial cells, and TGF β -related signalling potentially also is a key regulator of EndMT^{27, 28}. Thus, we explored the effects of *Trp53* and *Tgfbr2* on radiation-induced EndMT. At 48 h post irradiation (hpi), small-interfering RNA (siRNA)-mediated *Trp53* silencing in human umbilical vein ECs (HUVECs) markedly inhibited irradiation-induced mRNA levels of *Snail1*, *Snail2*, and *Zeb2*, which encode transcription factors implicated in EndMT²⁹, compared to control siRNA-treated cells, whereas *Tgfbr2* knockdown increased these levels (Supplementary Fig. 1a, b). Accordingly, overexpression of *Trp53*, but not *Tgfbr2*, augmented irradiation-induced increases in the EndMT markers filamentous actin, vimentin, and SMA, while reversing irradiation-inhibited CD31 levels (Supplementary Fig. 1c, d). Pericytes significantly restored the impaired tubule formation seen in irradiated ECs (compared to non-irradiated ECs), but not in *Trp53*-knockdown cells where pericyte recruitment was inhibited (Supplementary Fig. 1e). In contrast, *Tgfbr2* knockdown significantly enhanced pericyte integration into irradiated EC complexes and recovered EC tubule formation (Supplementary Fig. 1e).

Discussion (p. 18): We have added references to provide justification for deleting *Tgfbr2* in mice, as follows: ‘In support of this hypothesis, despite the essential effects of the TGF β pathway on cancer progression, inhibiting TGF β signalling in specific micro-environmental niches has produced conflicting results⁴². For example, suppressing TGF β signalling in fibroblasts promoted tumour progression^{43, 44}, whereas stromal TGF β R2 expression decreased as tumours progressed towards invasiveness⁴⁵’.

2) The investigators show that irradiation increases the number of CD31+ α SMA+ and NG2+ α SMA+ cells in a syngeneic tumor implant model, and they report that endothelial knockdown/knockout of p53 and TGF β R2 modulate this effect. They conclude that the changes in the absolute and relative numbers of these cell populations are due to EndMT, but no lineage tracing experiments were performed. A lineage tracing experiment is necessary to prove that EndMT contributes to changes in the tumor pericyte population.

Response: To address the reviewer’s apt comment, we have provided additional lineage-tracing data in Supplementary Fig. 4c. In addition, we have added the following text in the Results section (p. 6): ‘We analyzed the EndMT and pericyte population in greater detail by endothelial lineage-tracking using

cells expressing tdTomato-labelled Cre. To this end, we generated EC-tdTomato and EC-tdTomato;p53KO mice (Tie2-Cre;tdTomato and Tie2-Cre;tdTomato Trp53^{flox/flox}, respectively) (Supplementary Fig. 4c i). Immunofluorescence data showed that tdTomato⁺SMA⁺ cells were significantly increased (>50%) at 7 dpi in tumours of EC-tdTomato mice, but not in tumours of EC-tdTomato-p53KO mice (Supplementary Fig. 4c ii, top and iii). However, we did not detect any tdTomato⁺NG2⁺ cells in EC-tdTomato mice (Supplementary Fig. 4c ii, bottom). During tumour regrowth after irradiation, the populations of NG2⁺SMA⁺ or NG2⁺SMA⁺ pericytes significantly increased around irradiated vessels. Based on these data, we cautiously suggest that NG2⁺SMA⁺ cells do not originate from ECs, but that NG2⁺SMA⁺ pericytes can be derived from ECs via EndMT⁺.

3) Why are the percentages of aSMA⁺CD31⁺/CD31⁺ and NG2⁺ CD31⁺/CD31⁺ cells different in the WT group in Figure 1e and 1g (~70%) vs Figure 2d and 2e (~30%)? The control groups should have similar results. Please explain the discrepancy.

Response: To resolve this discrepancy, we have examined the proliferation of isolated primary KP cells by FACS analysis. As described in the Results section (p. 7), we used isolated primary KP cells at passage 4 or less to maintain the cellular characteristics of spontaneous lung tumour. We suggest that the discrepancy in the WT tumour-growth rates in Fig. 1b and 2b was related to the different percentages of aSMA⁺CD31⁺/CD31⁺ and NG2⁺CD31⁺/CD31⁺ cells (used to correct the formation of vessels with NG2⁺ pericytes). After irradiation, the proliferation rate of the primary KP cells used in the tumour-growth experiment shown in Fig. 2b was lower than that of the KP cells represented in Fig. 1b (Supplementary Fig. 5k), which may have caused the different growth rates of WT tumours in Fig. 1b and 2b.

Additionally, considering this experimental difference, we have added tumour-growth and immunofluorescence analysis data for TGFβR2KD mice (Fig. 2g) from a tumour-growth experiment performed with EC-p53KO;TGFβR2KD mice.

4) The central mechanistic role of osteopontin needs to be further proven. The investigators must show that osteopontin inhibition can rescue the phenotype of Tgfbr-2 conditional knockout mice.

Response: To address the reviewer's comment, we have revised the text in the Results section (p. 12) as follows: 'To better define the role of OPN secretion during tumour regrowth after radiotherapy, we examined the effects of a neutralizing anti-OPN antibody on EC-TGFβR2KD tumours. Notably, in EC-TGFβR2KD mice, the neutralizing anti-OPN antibody reduced tumour growth after irradiation, compared to control IgG (Supplementary Fig. 12a). However, compared to control IgG, the neutralizing anti-OPN antibody markedly reduced the formation of OPN⁺SMA⁺ vessels (from >60% to

10%) and proliferative CD44v6⁺ cells in EC-TGFβR2KD tumours at 21 dpi, and the anti-OPN antibody showed similar effects in WT tumours (Supplementary Fig. 12b, c)'.

These findings suggest that regulation of OPN secretion during EndMT may provide a new strategy to enhance the efficacy of radiation therapy.

5) In Fig 4g the investigators report that irradiation induces an increase in percentage of OPN+CD31+/CD31+ cells, which is blocked by p53KO. However, this percentage does not demonstrate that the population of cells that have undergone EndMT are the source of OPN. The investigators must demonstrate that irradiation induces changes in osteopontin expression occur in cells that have undergone EndMT.

Response: We thank the reviewer for this insightful comment. In Fig. 4g and Supplementary Fig. 11a, we have added representative images and the percentages of OPN⁺CD31⁺SMA⁺/CD31⁺ vessels to show that vessels that had undergone EndMT secreted OPN. In addition, Supplementary Fig. 11b shows that the number of OPN⁺SMA⁺ vessels increased significantly by 7 dpi.

6) Several markers of cancer stem cells have been proposed. The authors justify the use of CD44v6+ given its mechanistic relationship with OPN. However, the investigators must demonstrate that the CD44v6+ cells display other markers of stem-ness to make the claim that OPN induces proliferation of cancer stem cells.

Response: As mentioned in the Results section (p. 12), immunofluorescence data showed that nuclear expression of several stemness markers (Oct-4/Sox-2/β-catenin) increased substantially in the OPN-induced CD44v6⁺ population under hypoxia, compared to irradiated KP cells (Supplementary Fig. 11d).

7) The investigators report that “Tumour EndMT-recruited abnormal SMA+NG+ [NG2+] pericytes were detected in >60% of tumour vessels in tissues of patients who underwent surgery following neoadjuvant or combined chemo-radiotherapy, compared to patients who did not receive radiotherapy.” First, this statement should be revised to simply report the findings without asserting that the observed findings are due to EndMT recruitment, as that cannot be proven using this approach. At best, their findings are correlative. Secondly, the data are presented as aSMA+CD31+/CD31+ cells and NG2+CD31+/CD31+ cells, and it is unclear if 60% of the pericytes are double-positive (aSMA+ and NG2+). Quantification should be provided.

Response: In agreement with the reviewer's apt comment, we have revised the text as follows: 'Tumour SMA⁺CD31⁺ vessels and coverage with SMA⁺NG2⁺ pericytes were correlated in >60% of tumour vessels in tissues of patients who underwent surgery following neoadjuvant or combined chemo-radiotherapy, compared to patients who did not receive radiotherapy (Fig. 7a)' (p. 15).

In addition, we have added a graph showing the percentage of vessels with SMA⁺NG2⁺ pericytes (Fig. 7a). The graph presented as NG2⁺CD31⁺/CD31⁺ cells was corrected as vessels with NG2⁺ pericytes (Fig. 7a).

8) The authors report that human lung cancer samples that had been irradiated had a higher proportion of SDF1⁺CD206⁺ (M2-type) macrophages. However, they do not quantify M1 type macrophages. The investigators should show relative percentages of M1/M2 macrophages to determine the degree of polarization.

Response: To show the relative percentages of M1/M2 macrophages and thereby determine the degree of polarization as kindly suggested by the reviewer, we have added data on SDF1⁺CD206⁺CD68⁺ M2 macrophages and SDF1⁻iNOS⁺CD68⁺ M1 macrophages in Fig. 7d and Supplementary Fig. 15.

We have revised the text in the Results section (p. 15) as follows: ‘SDF-1⁻iNOS⁺CD68⁺ M1 macrophages were hardly detected (<10%), although they were detected more in irradiated than in non-irradiated tissues (Fig. 7d, Supplementary Fig. 15)’.

These data indicate the possibility of SDF⁺ M2 macrophage polarization in irradiated human tumours.

9) The analysis of the pericytes recruitment as a consequence of the irradiation is not clear. The investigators refer to recruitment in terms of appearance of NG2⁺ cells in the tumor vasculature and they presented a quantification of the CD31+NG2⁺ double positive cells. This interpretation is misleading as pericytes are physically in contact with endothelial cells but do not express endothelial markers.

Are the authors truly looking at the pericytes coverage of the tumor vasculature or is NG2 a mesenchymal marker expressed by the CD31⁺ endothelial cells as consequence of the activation of the EndMT program?

Response: We apologize for the confusing representation of the results. In this study, we examined the pericyte coverage of tumour vasculature (not co-expression of CD31 and NG2) as a measure of EndMT program activation. We have corrected this as ‘% vessels with NG2⁺ pericytes ’ in the figures 1h, 2e, 2i and 7a and these figure legends.

As supporting evidence, as described in our response to comment #2 regarding lineage tracing, we did not detect any tdTomato⁺NG2⁺ cells in irradiated tumours of EC-tdTomato mice with the endothelial lineage-tracking system, but tdTomato⁺SMA⁺ cells were significantly increased compared to in non-irradiated tumours (Fig. S4c).

10) The transition from one mechanism to another, as it relates to the tumor biology the authors are focusing on is sometimes not clear. For example, was CD44v6 the only CSCs marker with a differential expression in the irradiated WT compared to the irradiated p53 KO tumors? The expression of EpCAM, CD133 and ALDH is presented only in the irradiated versus non-irradiated WT tumors, and no information is provided

regarding their status in the p53 KO model. Therefore is not clear whether CD44v6 was chosen because among the CSCs marker it is the most upregulated in the irradiated WT tumors, or if because it is the only marker whose expression changes in the WT IR compared to p53 KO IR.

Response: We thank the reviewer for this apt comment, and we apologize for having failed to motivate our choice aptly in the text. We have added data showing the expression of EpCAM, CD133, CD44, and ALDH in WT and EC-p53KO mice after radiotherapy (Supplementary Fig. 7a, b).

The text in the Results section (p. 8) was revised as follows: ‘In radioresistant CSCs during tumour regrowth, aldehyde dehydrogenase⁺, CD44⁺, CD133⁺, and epithelial adhesion molecule⁺ lesion areas increased by 11%, 19%, 24%, and 4% respectively, in irradiated versus control tumours, which showed no significant difference compared to irradiated EC-p53KO tumours (Supplementary Fig. 7a, b). However, CD44v6⁺ areas increased by >50% in WT tumours but remained at 29% in EC-p53KO tumours after radiotherapy (Fig. 3a, Supplementary Fig. 7a)’.

Among the CSC markers, the CD44v6⁺ cell population was the most increased after radiotherapy. Also, we hypothesize that CD44v6 may be the only marker affected by regulating radiation-induced tumour EndMT, compared to other CSC markers.

11) Is the increase in hypoxia observed in the p53 KO tumors caused by an increase in the vascular leakage due to the loss of pericytes coverage when EndMT is inhibited?

Response: We thank the reviewer for this interesting question. We have examined vascular leakage using FITC-dextran, as shown in Supplementary Fig. 4g.

The text in the Results section (p. 6) was revised to: ‘Furthermore, significant leakage of FITC-dextran (indicative of intratumoural leakage) was observed in irradiated EC-p53KO compared to irradiated WT tumours (Supplementary Fig. 4g)’.

These data indicate that vascular leakage was increased in irradiated EC-p53KO tumours.

In addition, the text in the Results section (p. 9) was revised to: ‘We hypothesise that EndMT inhibition in irradiated EC-p53KO tumours resulted in a loss of pericyte coverage and subsequent vascular leakage, resulting in increased tumour hypoxia’.

12) There is no data demonstrating that p53 was successfully knock-down/knock-out in the Tie2-Cre;p53 mouse model.

Response: In agreement with the reviewer’s comment, the text in the Results section (p. 5) was revised to: ‘Trp53 mRNA⁻VE-cadherin⁺ cells were dominant in EC-p53KO, but not wild-type (WT) tumours,

indicating that p53 was successfully knocked out in tumour ECs of EC-p53KO mice (Supplementary Fig. 3a)'.

13) There are some discrepancies in the growth curves of KP WT tumors across the different mouse models (compare Fig. 1b, 2b and 2g). In particular, the growth rate of the WT tumor in Fig.2b seems to be considerably slower compared to the other, thus highly affecting the conclusion that tumors in the TGFβRII KD model growth more than the WT. Please clarify.

Response: As explained in our response to comment #3, after irradiation, the proliferation rate of the primary KP cells used in the tumour-growth experiment shown in Fig. 2b was lower than that of the KP cells represented in Fig. 1b (Supplementary Fig. 5k), which may have caused the different growth rates of WT tumours in Fig. 1 and 2.

Considering the experimental difference, we have added data for TGFβR2KD mice, which were analysed in a tumour-growth experiment with EC-p53KO;TGFβR2KD mice (Fig. 2i). These data showed that EC-Tgfr2 knockdown significantly enhanced tumour growth post-irradiation.

14) In Figure 4c/d, the authors supported the result regarding the OPN transcript levels in the HUVEC (4c) with a cytokine array performed on CM media from a completely different endothelial cells model (human pulmonary microvascular endothelial cells). What is the level of secreted OPN in the HUVEC under all the different conditions presented in Fig.4c?

Response: We performed OPN ELISAs with conditioned media from HUVECs transfected with siRNAs against Trp53, Tgfr2, or Trp53+Tgfr2 (Supplementary Fig. 10c).

We have revised the text in the Results section (p. 11) as follows: 'Coincident with *OPN* upregulation, secreted OPN increased in control and TGFβR2-deficient ECs at 5 dpi, but significantly decreased after *Trp53* knockdown in conditioned HUVEC medium (Supplementary Fig. 10c)'.

15) Statistical analysis is missing in all tumor growth curves.

Response: We have added statistics for all tumour-growth curves.

16) Please add quantification of the staining in the following figures: 6b, S3d, S4h.

Response: We have added quantitative data in Fig. 6b and Supplementary Fig. 3d, 4h (Fig. 6b and Supplementary Fig. 4e, 5j in the revised manuscript).

17) Figure S1c: what are the CD31+ area and vessel diameter in the regression stage?

Response: We have added the CD31⁺ area and vessel diameters observed during the regression stage in Supplementary Fig. 1c.

18) Regarding the data shown in Figure S4a-b, the authors stated that the levels of pSmad2/3 and EndMT after IR or TGFb1 treatment differentially changed depending on which TGFb receptor is knock-down. However only pSmad2/3 levels are shown in the figure, no data on the EndMT phenotype.

Response: We thank the reviewer for pointing this out. We have added immunofluorescence images and quantitative data on the EndMT phenotype in Supplementary Fig. 5c, d.

The text in the Results section (p. 7) has been revised as: ‘Radiation- or TGFβ1-induced increases in EndMT markers (filamentous actin and FSP1) were decreased in TGFβR1-depleted HUVECs, but were markedly increased in TGFβR2-depleted cells (Supplementary Fig. 5c, d)’.

Reviewer #2 (Remarks to the Author):

NCOMMS-17-33270

Choi et al. “Tumor-vasculature development via the endothelial-to-mesenchymal transition after radiotherapy controls CD44v6⁺ cancer cell and macrophage polarization “

The authors study the effect of 20Gy of radiation in mouse preclinical models (one colon cancer and one lung cancer KRAS p53) of cancer on the associated tumor vasculature and then on a cancer stem cell subpopulation. Their major findings involve detection of endothelial cell mesenchymal transition after radiation and the effect this has on protecting a hypoxic cancer stem cell subpopulation. As part of this they use knockout models (TP53 and TGFRB2 receptor) to modulate effects. As part of this they study the presence of M2 (immune suppressive) and M1 (immune stimulatory) macrophages after radiation and the mechanisms involved (including osteopontin secretion). They end by providing data on immune histochemical analyses of human lung cancers for various endothelial cell, macrophage, and tumor stem cell markers with and without radiation treatment. They conclude: “Our findings suggest that targeting tumour EndMT might enhance radiotherapy efficacy by inhibiting the re-activation of dormant hypoxic CSCs and promoting anti-tumour immune responses. “

Comments to the authors:

The manuscript is reviewed in the context for urgent need to understand the effects of radiation on tumor treatment and potential immuno-stimulatory vs. immuno-inhibitory and cancer stem cell ablative vs. protective effect of radiation and potential ways to develop new therapies. This is particularly important given the recent advances of using stereotactic ablative radiation therapy for lung cancer including its role in local

consolidation for oligometastatic disease.

All of the experiments are technically well done and presented in great detail. There are several issues the authors need to address. Of all of the issues, the most important is the information required to understand the studies of the human tissues.

1. The way the paper is presented, as a whole is very difficult to read to identify their key points. They should include some kind of summary schema of their current findings that would provide a road map for integrating their findings. As part of this they can indicate what they think the key biomarkers would be and potential therapeutic targets.

Response: We have added a schematic summary of our findings in Fig. 7e, with the following legend (p. 34, 35): ‘Model for TRP53-regulated irradiation-induced EndMT and tumour vasculature. Radiation-induced tumour EndMT causes aggressive tumour vasculature, recruiting abnormal α SMA⁺ or NG2⁺ α SMA⁺ pericytes. Firstly, vascular ECs occur in EndMT secreted OPN, which trigger the proliferation of radioresistant dormant hypoxic CD44v6⁺ CSCs with metastatic potential. Secondly, EndMT cells overexpressing CXCR4 serve as a reservoir for M2 macrophages polarized from SDF-1-positive monocytic cells. These phenomena synergistically affect aggressive tumour regrowth after radiotherapy. However, endothelial TRP53 deletion inhibits radiation-reduced EndMT and aberrant tumour vasculature. Subsequently, inhibited OPN secretion and CXCR4 expression can synergistically inhibit tumour regrowth after radiotherapy. Our findings suggest that targeting radiation-tumour EndMT may enhance radiotherapy efficacy both by inhibiting the reactivation of dormant CSCs and by promoting radiation-antitumor immune responses, representing a potentially viable, new therapy’.

The following text was added to the Discussion section (p. 16): ‘Based on our findings, we propose a model for TRP53-regulated radiation-induced EndMT and tumour vasculature as illustrated in Fig. 7e’.

2. All of their animal experiments appear to be done with 20 Gy of radiation (a very high dose). However, the methods provide no indication of how this was delivered or the fraction(s) used. While this can easily be corrected in the Methods I found the omission of this basic piece of information to be very serious.

Because of the use fractionated vs. stereotactic radiation therapy in the treatment of human lung cancer, it would be very important to know if the fractionation of the radiotherapy influences any of their key results.

Thus, standard doses to patients are given in 1.5-2 Gy fractions vs. very high dose single fraction ablative radiation therapy. Is there a difference in the EndMT? Obviously we need to know this whatever the answer is.

Response: In response to this apt comment, we have described the radiation delivery (which involved a single 20-Gy dose using the X-RAD 320 platform) in more detail in the Methods section (p. 22), the supplemental experimental procedures (p. 4), and in all figure legends.

The text in the Results section (p. 10) was modified as follows: ‘To examine whether fractionated radiotherapy influences radiation-tumour EndMT, we irradiated WT and EC-p53KO tumours with 30 Gy in six fractions (Supplementary Fig. 9a). We observed no difference in tumour growth and lung metastasis between WT and EC-p53KO (Supplementary Fig. 9b-d). However, the fractionated irradiation-induced increase in the SMA⁺CD31⁺ population in WT tumours was significantly inhibited in EC-p53KO tumours, whereas the pimonidazole-staining intensity and the population of Ki67⁺CD44v6⁺ cancer cells in hypoxic areas were not different between WT and EC-p53KO tumours (Supplementary Fig. 9e, f). In addition, we examined the effect of 20-Gy exposure in daily 2-Gy fractions in WT tumours and EC-p53KO mice from systemic tamoxifen-mediated-specific Cre recombination in the VE-cadherin promoter (Supplementary Fig. 9g-k). After therapy with daily 2-Gy fractions, WT tumour growth was significantly higher than that observed with a single dose of 20 Gy. However, with fractionated irradiation, EndMT occurrence and hypoxic staining density in WT tumours significantly decreased, compared to a single high dose (Supplementary Fig. 9j, k). Coincident with WT tumour growth, the population of proliferative CD44v6⁺ cancer cells in hypoxic areas increased more with 2-Gy fractions than with a single 20-Gy dose (Supplementary Fig. 9k). Moreover, no difference in tumour growth and the population of Ki67⁺CD44v6⁺ cancer cells occurred in the hypoxic areas in WT and EC-p53KO mice after ten daily 2-Gy fractions, even though EC-p53KO significantly reduced EndMT, compared to WT’.

These results suggested that fractionated radiotherapy caused less EndMT than single high-dose radiation therapy, whereas fractionated radiotherapy increased CD44v6⁺ CSC proliferation more than single high-dose radiation. Thus, in fractionated radiotherapy, EndMT inhibition cannot overcome proliferation of cancer cells, resulting in no significant difference in tumour growth. We hypothesize that EndMT-targeting strategies may be more effective in single high-dose radiation therapy than in conventional fractionated radiotherapy.

3. Some of the most important data are those from patients in Figure 7 and Supplemental Table 1. However, in reading over the patient numbers and methods I have no idea which data are used in Figure 7 and how the overall two datasets were used for the studies. I cannot stress strongly enough to the authors that I was very impressed with their work, but when I dug into the details about the human studies I was very disappointed by the information they provided. While I hope they can provide this key information, its lack in the manuscript at this point was a very big negative for me. I suspect this is the Origene N = 13 patient dataset but, if so, what data were generated from the Severance Hospital data set. In addition, in the Methods they discuss 27 tumors of which 3 received radiation therapy. In addition, from Table S1 it is clear the patients received chemotherapy with the radiation therapy so which treatment is responsible is, of course, clouded. Now such samples are hard to come by, but it is important for the authors to address this limitation in the discussion.

Response: We apologize that the original text was confusing regarding the human studies. We have addressed this by providing clearer explanations of the human studies (p. 15 in the Results section) and the human tissue specimens (Methods section, the legend of Fig. 7, and Supplementary Table 1).

Also, we need to know the comparison demographics of the tumors that did and did not receive radiation (such as gender, histology, smoking status, stage, and if possible oncogenotype). Were the two groups comparable?

Response: We apologize that this information was lacking. We have added clinicopathologic data in Supplementary Table 1 and provided comparative demographics in Supplementary Table 2. However, there were no other comparable factors among clinicopathologic characteristics.

Finally, the mouse studies were with 20 Gy while the patient studies were with fractionated radiation therapy. Thus, it is very important the authors provide preclinical data that can be matched with the treatment given to their patient specimens.

Response: As described in our response to comment #2, we have added data from mouse studies indicating that regulating EndMT with fractionated radiation therapy was not efficient in inhibiting tumour growth because CSC proliferation was not inhibited, compared to that observed after a single high dose (p. 10).

We have modified the text in the Results section (p. 15) as follows: ‘The patients who received radiotherapy received fractionated doses, not a single high dose, and our mouse studies (Supplementary Fig. 9) had shown significant EndMT after fractionated radiotherapy; thus, the occurrence of EndMT and subsequent phenomena in human tissues can support the clinical relevance of our data.’

We have modified the text in the Discussion section (p. 20) as follows: ‘Coincidentally, in this study, targeting EndMT more efficiently inhibited tumour regrowth after a single high dose of radiation than after fractionated radiotherapy. In fractionated radiotherapy, tumour EndMT was significantly inhibited by EC-p53KO; however, CD44v6⁺ CSC proliferation and tumour growth were not inhibited. Nevertheless, we cautiously suggest that in fractionated radiotherapy, targeting EndMT may enhance therapeutic efficacy when combined with a strategy to inhibit CSC proliferation.’

Reviewer #3 (Remarks to the Author):

Choi et al. present an interesting story on the effect of irradiation on p53-dependent endothelial to mesenchymal transition. the experiments are elegantly conducted and well controlled.

Several points in the manuscript are less well explored while others might need some clarification.

General:

The biggest concern from the reviewer is the lack of mechanisms shown in the manuscript. The authors show that deletion of p53 in endothelial cells results in increased tumor control reducing of cell death accompanied by reduced EndMT. EndMT in turn is inhibited via hypoxia (potentially due to reduced cell death) and M2 macrophages. Loss of p53 results in ablation of SDF1 and reduced macrophage recruitment. in a beautiful experiment in figure 5f the authors prove the M2 macrophages drive EndMT directly but not through which mechanisms. This should be a small and testable list of possibilities and thus should be explored to show mechanism and provide therapeutic opportunities. the manuscript would benefit greatly from a summary schematic

Response: In agreement with the reviewer's helpful suggestion, to better clarify our findings, we have added a schematic summary in Fig. 7e.

The following legend was added for Fig. 7e: 'Model for TRP53-regulated irradiation-induced EndMT and tumour vasculature. Radiation-induced tumour EndMT causes aggressive tumour vasculature, recruiting abnormal α SMA⁺ or NG2⁺ α SMA⁺ pericytes. Firstly, vascular ECs occur in EndMT secreted OPN, which trigger the proliferation of radioresistant dormant hypoxic CD44v6⁺ CSCs with metastatic potential. Secondly, EndMT cells overexpressing CXCR4 serve as a reservoir for M2 macrophages polarized from SDF-1-positive monocytic cells. These phenomena synergistically affect aggressive tumour regrowth after radiotherapy. However, endothelial TRP53 deletion inhibits radiation-reduced EndMT and aberrant tumour vasculature. Subsequently, inhibited OPN secretion and CXCR4 expression can synergistically inhibit tumour regrowth after radiotherapy'.

Figure 1: when were the tumors irradiated? Data showing decreased cell death in p53-KO mice should be included in the main figure.

Response: We thank the reviewer for pointing this out. We have indicated the irradiation time points with arrows in Fig. 1 and added an explanation in the legend of Fig. 1. We have also added data showing the necrotic areas of WT and EC-p53KO mice in Fig. 1d.

The text in the Result section (p. 5) was modified as follows: ‘7 days post irradiation (dpi), necrotic areas and the population of apoptotic cells were increased more in p53KO than in WT tumours (Fig. 1d, Supplementary Fig. 3b)’.

Figure 2: d/e/i in i) the authors compare double KO with WT (60% and 40% SMA and NG2+ cells) while in d) WT is 40% and in e) Wt is 30%. This could be due to experimental difference but should this be the case the authors need to include single KO in figure i.

Response: Considering the experimental differences, we have added data for TGFβR2KD mice (Fig. 2i) based on a tumour-growth experiment conducted with EC-p53KO;TGFβR2KD mice.

The text in the Results section (p. 8) was modified as follows: ‘Following 20-Gy irradiation, tumour growth was significantly delayed in EC-p53KD/KO;TGFβR2KD versus EC-TGFβR2KD mice (Fig. 2g, h). Vessel numbers with SMA⁺NG2⁺ pericytes decreased markedly after irradiation in EC-p53KO;TGFβR2KD versus EC-TGFβR2KD tumours, but increased in TGFβR2KD versus WT tumours (Fig. 2i)’.

Figure 3: e is missing f exists twice.

Response: We thank the reviewer for pointing this out; we have corrected this.

Figure 5: The authors solely focus on macrophages but other immune cells could be affected likewise. Are T cell numbers different? other MHCII antigen presenting cells?

Response: Per the reviewer’s insightful inquiry, we have examined whether other immune cells could be affected using immunofluorescence; data are shown in Supplementary Fig. 14.

We have modified the text in the Results section (p. 14) as follows: ‘Next, we explored the responses of other immune cells in WT and EC-p53KO tumours after radiotherapy. Immunofluorescence data showed that at 7 dpi, the population of granzyme B (GZMB)⁺CD8⁺ cytotoxic T cells was increased in EC-p53KO, but not WT tumours, compared to non-irradiated tumours. The GZMB⁺CD8⁺ population was significantly increased in irradiated EC-p53KO tumours (Supplementary Fig. 14a). The population of CD4⁺Foxp3⁺ regulatory T cells increased in WT tumours after radiation, and no difference was found between irradiated WT and EC-p53KO tumours (Supplementary Fig. 14b). Additionally, at 7 dpi, the population of MHCII⁺ antigen-presenting cells was significantly increased in EC-p53KO, but not WT tumours, compared to non-irradiated tumours (Supplementary Fig. 14c)’.

Taken together, our results suggests that after radiation therapy, EC-p53KO in tumours may regulate the population of immune cells, including cytotoxic T cells and MHCII⁺ antigen-presenting cells, but whether these phenomena were directly affected by regulating EndMT requires further study.

Reviewers' Comments:

Reviewer #1:

Remarks to the Author:

The authors have addressed all comments satisfactorily.

Minor points:

- Figure S1a-b: the authors should include gene expression analysis of the non-irradiated HUVEC in order to appreciate the radiation-induced increased expression of the indicated EndMT genes.
- The authors referred to the NG2–aSMA+ cells increasing around irradiated vessels as pericytes. NG2 positivity is often used to discern pericytes from fibroblasts, not the lack thereof. Consider qualifying these statements (possibly refer to these cells simply as aSMA+ mesenchymal cells as, most likely, they are derived from endothelial cells through EndMT as shown by the authors' data?)

Reviewer #2:

Remarks to the Author:

The authors have responded to all of the reviewers' comments including providing a substantial amount of additional data.

Reviewer #3:

Remarks to the Author:

All concerns of the reviewer have been addressed.

Responses to the Reviewers' comments

We thank the reviewers for the insightful comments. Please find our point-by-point responses below.

Reviewer #1 (Remarks to the Author):

Minor points:

- Figure S1a-b: the authors should include gene expression analysis of the non-irradiated HUVEC in order to appreciate the radiation-induced increased expression of the indicated EndMT genes.

Response: We have added data showing the gene expression of the non-irradiated HUVEC in Supplementary Fig 1a and b.

- The authors referred to the NG2-aSMA+ cells increasing around irradiated vessels as pericytes. NG2 positivity is often used to discern pericytes from fibroblasts, not the lack thereof. Consider qualifying these statements (possibly refer to these cells simply as aSMA+ mesenchymal cells as, most likely, they are derived from endothelial cells through EndMT as shown by the authors' data)

Response: In agreement with the reviewer's apt comment, we have revised NG2⁺SMA⁺ cells as SMA⁺ mesenchymal cells (page 6)